# The Association between Corporate Social Responsibility, Employee Performance, and Turnover Intention Moderated by Organizational Identification and Commitment

**Mohammad Alnehabi** [1],* and **Al-Baraa Abdulrahman Al-Mekhlafi** [2]

1   Department of Business Administration, College of Economics and Administrative Sciences,
    Imam Mohammad Ibn Saud Islamic University (IMSIU), Riyadh 11623, Saudi Arabia
2   Faculty of Leadership and Management, Universiti Sains Islam Malaysia (USIM), Nilai 71800, Malaysia;
    albaraa901@gmail.com
*   Correspondence: mohalnehabi@gmail.com

**Abstract:** Corporate social responsibility (CSR) holds increasing significance within Saudi Arabia's banking sector. By adopting responsible and sustainable practices, banks can not only enhance their financial performance but also bolster the trust and loyalty of their customers. The sector recognizes that high turnover rates and subpar performance can lead to elevated costs and reduced trust in the bank's services. Consequently, this study aims to investigate how organizational identification and commitment mediate the relationship between CSR, employee performance (EP), and turnover intention (TI). Following a survey with 550 employees, the structural equation modelling technique was applied to test the study's model and complex relationships. The study assessed 12 hypotheses, 8 of which represented direct relationships, while the remaining 4 explained the mechanisms of the mediating relationships. All of these hypotheses show significant relationships. All variables explained the variance of EP by 42% while explaining the variance of TI by 28%, which had a moderate effect on the dependent variables. The model indicates that values are well constructed and that the model has predictive relevance due to $Q2$ being above 0. The study's findings demonstrate that organizational identification and commitment channel the link between corporate social responsibility and employee performance and turnover intention. The study underscores the significance of CSR, organizational identification, and commitment in the Saudi Arabian banking sector. It provides valuable insights for banks to enhance employee performance, reduce turnover intention, and strengthen corporate social responsibility initiatives.

**Keywords:** corporate social responsibility; employee performance; organizational identification; organizational commitment; turnover intention

## 1. Introduction

The Saudi banking sector, a cornerstone of the nation's economy, faces a critical challenge: the issue of employee turnover intention. This phenomenon, often accompanied by a decline in employee performance, not only threatens the industry's stability but also carries substantial economic implications for Saudi Arabia's growth [1]. High turnover rates incur substantial costs, strain employee morale, and impede overall productivity, ultimately impacting customer service quality [2–4].

In this context, diminishing employee performance in the banking sector can have severe repercussions, affecting profitability, customer trust, and organizational efficiency [5,6]. Furthermore, it can lead to increased turnover rates in the banking sector. Only 50% of banking industry employees are highly engaged, with 35% being a retention risk. Organizations with highly engaged workforces report decreased turnover. This can make it more challenging for financial institutions to find, hire, and keep talented people. It may also cause customers to feel less committed to the bank, which might have an effect on the institution's bottom line [7].

It is essential to foster organizational commitment and identification among employees to address the critical challenge of employee turnover intention and performance in the Saudi banking sector. Organizational commitment, encompassing emotional attachment and engagement, plays a significant role. Organizational identification, which pertains to a sense of belonging, is equally essential. Corporate social responsibility directed toward employees can bolster their identification and commitment to the organization, ultimately benefiting the bank. These factors profoundly impact profitability and competitive positioning [8,9]. Hence, fostering these attributes among employees is pivotal.

Numerous studies have investigated various factors that influence the Saudi banking sector. These factors include rewards and motivation [10], occupational stress, psychological stress, and burnout [11,12]; internal marketing [13], human resources management strategy [14,15], employee job satisfaction, and human resource management practices Cherif [16,17]; and emotional intelligence [18]. However, there is a paucity of research examining the role of organizational identification (OI) and commitment (OC) as mediators of the relationship between corporate social responsibility (CSR), employee performance (EP), and turnover intention (TI). Due to the scarcity of research exploring the mediating roles of OI and OC in these relationships, this study proposes that employing OC and OI as mediators will facilitate a deeper understanding of the link between the independent variable (CSR) and the dependent variables (EP and TI) in this specific context. The mechanisms through which OC and OI impact TI and EP have largely remained uncharted territory, especially within the Saudi Arabian banking sector, and have not garnered sufficient attention. Consequently, an urgent imperative exists to broaden the theoretical and empirical underpinnings to grasp the essence of the relationship between CSR, EP, TI, and the mediating factors of OI and OC. Thus, this leads us to the central research question: How do organizational identification (OI) and commitment (OC) mediate the relationship between corporate social responsibility, employee performance, and turnover intention among employees in the banking sector?

## 2. Literature Review

### 2.1. Relationship between the CSR and EP

For the sake of this discussion, "employee performance" refers to the duties and tasks that workers are paid to complete. It includes actions that fulfill the job description's requirements [19–21]. Story and Neves Story and Neves [22] found that CSR influenced productivity levels of workers. In particular, workers' in-role performance increased when they believed the company cared about important issues. They also discovered that internal performance, emotional commitment, and work satisfaction had clear links to outward CSR [23]. According to Edwards and Kudret Edwards and Kudret [24], CSR attitudes affect employee performance, particularly perceptions of fairness and dedication.

Having established the link between CSR and employee performance, it is now essential to bridge these findings with the overarching objective of our research in order to investigate the impact of CSR initiatives on workplace productivity. Theories of social identities—individuals identify with a social entity when they feel a connection to it, as explained by Tajfel and Turner Tajfel and Turner [25]. Employees connect more strongly with their employer when the company demonstrates social and ecological responsibility in its actions and communication. When employees see alignment between their values and the company's activities, they are more likely to identify with the organization and exert greater effort. Previous research has affirmed the positive link between organizational identification and performance [26].

A Saudi Arabian service industry study found that CSR initiatives enhance company performance. Typical CSR methods that benefit organizations were identified in this research [27]. Another study in the hotel and tourism sector in Hail, Saudi Arabia highlighted how external CSR actions related to the local community during the COVID-19 pandemic significantly impacted employee performance. The findings underscored the link between corporate social responsibility and increased workplace productivity [28].

However, the banking industry in Saudi Arabia has yet to look into this connection. Given this background, we now suggest the following hypothesis:

**Hypothesis 1:** *A positive relationship exists between corporate social responsibility and employee performance.*

### 2.2. Relationship between the CSR and OI

Employee engagement in CSR initiatives has the potential to enhance employee satisfaction through these activities, consequently leading to an increase in the level of organizational identification (OI) [29,30]. When employees perceive their companies as socially responsible organizations, their sense of belongingness can be improved, positively impacting their self-concepts [31].

Employees have been found to have a more positive attitude toward their employers if they have a favorable impression of the company's character, values, and social responsibilities [32,33]. Top business school MBA graduates are prepared to take a pay cut to work for businesses with a strong corporate social responsibility (CSR) presence, according to research by Montgomery and Ramus Montgomery and Ramus [34]. Another study that found a positive correlation between CSR and OI was done by Jacinto and Carvalho Jacinto and Carvalho [35], who examined the effect of CSR activities on workers' organizational identification (OI) and work habits. Del Rosario Del Rosario [36] and Brammer Brammer [37] also found a favorable and direct relationship between CSR and OI.

Compared to other businesses, socially responsible companies have a better corporate reputation (CR), making it more appealing for potential employees to seek employment with them [37,38]. Unfortunately, the link between CSR and OI has not been experimentally investigated in the Saudi Arabian context despite abundant data in other contexts such as Belgium, Spain, and Korea. Therefore, it is important to look into whether or not this relationship exists in the banking industry of Saudi Arabia in order to compete with or supplement the established partnerships in other contexts. Therefore, we postulate the following hypothesis:

**Hypothesis 2:** *There is a positive relationship between corporate social responsibility and organizational identification.*

### 2.3. Relationship between the CSR and OC

Robust and reliable employee–employer relationships serve as the foundation for thriving corporations. In order to achieve sustainable success in today's highly competitive business landscape, companies must cultivate an environment that attracts and retains talented and dedicated employees [39,40].

CSR is considered a crucial factor influencing employees' behaviors and attitudes in the workplace, including organizational commitment (OC) [33,41,42]. Existing empirical research has consistently supported a positive relationship between CSR and OC [41,43–47]. Supporting these findings with social identity theory, Turker Turker [48] investigated the CSR-OC relationship and established that CSR significantly predicts OC. Thus, substantial evidence exists to expect a positive relationship between CSR and OC.

However, despite the abundance of evidence from various contexts, such as Korea, India, Turkey, and Tunisia, the relationship between CSR and OC has not been empirically tested in Saudi Arabia's banking sector. It becomes crucial to investigate further to determine whether this relationship exists in Saudi Arabia and to either challenge or complement the established relationships in other contexts. Therefore, the following hypothesis is proposed for testing:

**Hypothesis 3:** *There is a positive relationship between corporate social responsibility and organizational commitment.*

*2.4. Relationship between the CSR and TI*

Several studies have examined how CSR affects workers' willingness to leave an organization. Corporate social responsibility and internal marketing have a considerable negative effect on turnover intentions in Tehran, according to research by Boudlaie Boudlaie [49]. Another study out of Taiwan found that when workers see CSR initiatives being implemented internally, they are more committed to their jobs and less inclined to go elsewhere for employment. However, the reverse may occur if workers become aware of corporate duplicity and become more likely to consider leaving their positions [50]. Individuals with a strong demand for status in the United States had a more positive mood and less turnover intention when the CSR effort benefited a temporary cause [51]. Studying the impact of CSR on worker turnover intentions in China required combining social identity theory and social exchange theory. Corporate social responsibility was found to have a negative effect on employees' desire to leave their current positions, with the effect being larger for internal CSR initiatives than for those focused on the broader community [52]. However, in Saudi Arabia, especially in the banking sector, the connection between CSR and turnover intention purpose has been ignored.

Therefore, it can be concluded that corporate social responsibility can have an impact on turnover intention among employees. When organizations prioritize CSR and implement it effectively, it can lead to increased dedication to work and lower turnover intention. As a consequence of this, the following hypothesis has been put up to be tested:

**Hypothesis 4:** *There is a negative relationship between corporate social responsibility and turnover intention.*

*2.5. Relationship between OC and EP*

The term "organizational commitment" describes a person's emotional investment and dedication to their workplace. It indicates that they are committed to the company and intend to stay for the long haul [53,54]. Research findings suggest that the relationship between organizational commitment and employee performance is more favorable and substantial for sales employees than non-sales employees [55]. Moreover, the connection between organizational commitment and job performance in collectivist cultures tends to be stronger than in individualistic cultures.

Context-specific research has also demonstrated the favorable correlation between organizational dedication and productivity. For instance, Suharto Suharto [56] observed that government workers in Lampung Regency had a strong correlation between organizational dedication and performance on the job. Darolia Darolia [57] found that organizational commitment positively affected the work performance of male skilled professionals. Organizational commitment moderated the relationship between overall job stress and performance [58]. In another study, empowering leadership by principals was linked to improved teacher job performance, with organizational commitment as a mediator, indicating that strong commitment led to enhanced performance [59].

Additionally, organizational commitment mediated the connection between job satisfaction and employee performance, with highly committed and satisfied employees demonstrating higher performance levels [60]. Putra Putra [61] noted that an ethical work atmosphere increases employee happiness, with organizational dedication having a stronger impact on productivity. This research's results highlighted the role of dedication to the company in motivating workers. However, research on this vital link in the Saudi Arabian banking industry is scarce. Therefore, the current study anticipates that organizational commitment will influence employee performance, leading to the following hypothesis:

**Hypothesis 5:** *There is a positive relationship between organizational commitment and employee performance.*

## 2.6. Relationship between OC and TI

Several fields, including healthcare, hospitality, and government, have investigated the connection between organizational dedication and the desire to leave. There was a negative correlation between burnout and the desire to leave the nursing profession among ICU nurses. However, this correlation was attenuated for nurses who reported high levels of organizational commitment [62]. Perceived organizational support was found to correlate positively with organizational commitment and negatively with turnover intention among hotel industry workers [63].

Similar results were reported in the high-end hospitality industry during the 2009 COVID-19 pandemic, where employment instability negatively impacted organizational commitment and, in turn, turnover intent [64]. This study demonstrated that in government agencies, insider status somewhat moderated the connection between loyalty to the company and a desire to leave. For both sexes, perceived insider status had a reduced indirect influence on organizational commitment and turnover intention [65]. Employees' intentions to leave the company were shown to be significantly inversely related to their level of commitment to PT. X Mojokerto, according to the study's findings [66]. The data summarized above imply that to prevent employee turnover, businesses should work to increase employee commitment to the company. However, the financial industry in Saudi Arabia has ignored this critical issue. This study, therefore, tests the hypothesis that a person's commitment to their present company would influence their likelihood of leaving.

**Hypothesis 6:** *There is a negative relationship between organizational commitment and turnover intention.*

## 2.7. Relationship between OI and EP

The term "organizational identification" describes the extent to which an organization's staff members feel a personal connection to and take pride in their workplace [67]. It is a mental state that can profoundly influence how individuals function within the workplace. This process involves employees assimilating elements of the company's identity as their own, often with the encouragement of management. Organizational identification is closely linked to in-role performance, encompassing all the work-related actions expected within one's formal job role [68].

Strong organizational identification drives employees to invest extra effort, value their membership, follow the rules, and prioritize in-role performance. This commitment stems from their belief in contributing to the organization's success and preserving its uniqueness, motivating exceptional task performance [69].

One research indicated that employees' job performance improved when they had a strong connection to the firm they worked for. Employee loyalty, in turn, is significantly impacted by internalization at work, which is connected to organization identity, significantly impacting employee job performance [70]. In another study, the link between authoritarian leadership and work performance and organization-directed citizenship behavior was shown to be mediated by relational identification. This indicates that workers are more likely to succeed under authoritarian management when they strongly feel an affiliation with their workgroup and its cultural norms [71]. Notably, there is a lack of studies that address this relationship in Saudi Arabia's banking sector. Consequently, based on the connections above between organizational identification and in-role performance, the study proposes the following hypothesis:

**Hypothesis 7:** *There is a positive relationship between organizational identification and employee performance.*

### 2.8. Relationship between OI and TI

The term "turnover intention" describes a worker's plans to quit their present employer [72]. Employees with a strong sense of belonging to their firm are less likely to be interested in leaving their current position [73]. According to the literature, several elements, including diversity management, organizational support, and a sense of personal connection with a supervisor, have been found to affect the correlation between organizational identification and the desire to leave [72,74]. Organizational citizenship behavior predicts turnover intention, and one study revealed a positive relationship between organizational support and identification [72].

According to research, salespersons' intention to leave an organization has also been linked to how closely they feel they identify with their manager. This indicates that an employee's identification with their direct supervisor can have an effect on the correlation between organizational identification and intention to leave [74]. The desire to leave a company was shown to be inversely related to the degree to which employees felt their superiors supported them. This indicates that workers who feel strongly connected to their firm are less inclined to consider leaving their current position [73].

Thus, the relationship between organizational identification and turnover intention is complex and can be influenced by various factors. However, the Saudi Arabia bank sector has not investigated this relationship. Therefore, this research will test this hypothesis in the bank sector.

**Hypothesis 8:** *There is a negative relationship between organizational identification and turnover intention.*

### 2.9. The Mediating Role of the OC and OI in the Relationship between CSR and EP, TI

2.9.1. Organizational Commitment

Several studies have used organizational commitment as a mediating variable to explore its relationship with other factors. For example, a study on the Mercury Hotel Banjarmasin found that work placement and discipline affected work morale, with organizational commitment as a mediating variable. The study also found that work discipline influenced work morale with mediation by organizational commitment [75]. Organizational commitment was used as a moderator in research at CV Makmur Jaya Abadi in Surabaya City to determine the effect of transformational leadership, organizational culture, and management control systems on employee performance. According to the results, transformational leadership significantly affects employee loyalty to the organization [76].

Additionally, the research looked at how factors like stress and interpersonal communication have a role in whether or not private hospitals retain employees. According to the findings, organizational dedication was a mediator between the two variables [77]. Yoon and Jeon Yoon and Jeon [78] found that the connection between small enterprises' market orientation, learning orientation, and corporate performance was partially mediated by organizational commitment.

Organizational citizenship behavior among faculty members at private universities was the subject of this study. Organizational commitment influences an organization's culture, leadership, justice, and employees' civic activity [79]. Similar research looked at how factors including a culture of openness to failure, free flow of information, autonomy at work, and equitable pay affected employees' propensity to take risks and innovate. Organizational commitment was the mediator between these variables and entrepreneurial activity in the workplace [80]. Finally, research examined the link between human resource development elements, professional advancement, and the desire to leave the organization. The results showed that organizational commitment mediated the connection between these elements and the likelihood of leaving one's current job [81].

In summary, numerous studies have used organizational commitment as a mediator, finding it influences work morale, employee loyalty, retention, corporate performance, organizational culture, and more in various contexts. It acts as a critical link between key

factors across different domains. These studies highlight the importance of organizational commitment as a mediator in understanding the relationship between various factors and organizational outcomes. Thus, the current study applies this organizational commitment to mediating the relationship between CSR and TI, EP among banking sector employees in Saudi Arabia, and we propose the hypotheses below:

**Hypothesis 9:** *There is a mediating role of organizational commitment in the relationship between corporate social responsibility and employee performance.*

**Hypothesis 10:** *There is a mediating role of organizational commitment in the relationship between corporate social responsibility and turnover intention.*

2.9.2. Organizational Identification

Several different research studies have investigated the role of organizational identification as a mediating variable. One study looked at the connection between perceived supervisor support and commitment to change, as well as the impact that organizational Identification had on the relationship between perceived supervisor support and the various components of change commitment. According to Zappalà Zappalà [82], the findings demonstrated that organizational identification completely mediates the continuous commitment to change and partially mediates the emotional and normative commitment to change.

In separate research, the researchers looked at organizational identification's role as a mediator between corporate social responsibility and organizational performance. According to the findings, organizational identification is a significant internal activity that CSR enhances to improve organizational performance [83]. A study with multiple levels looked at the relationship between the promotion climate of corporate social responsibility (CSR) at the team level, the level of work satisfaction experienced by team members, and the role that organizational identification played as a mediator at the individual level. According to the findings, organizational identification partially mediates the connection between the team CSR promotion environment and employees' workplace satisfaction levels [84].

It can be concluded that the research demonstrates that organizational identification acts as a mediator, fully mediating continuous commitment to change and partially mediating emotional and normative commitment in relation to perceived supervisor support. Moreover, it enhances organizational performance through CSR and partially mediates the link between team-level CSR promotion and individual employee workplace satisfaction. Therefore, these studies suggest that organizational identification is an important internal process that can mediate the relationship between various factors and outcomes, such as CSR, job satisfaction, career satisfaction, and organizational commitment. Therefore, the current study will apply this variable as mediating in the relationship between CSR and TI, EP among banking sector employees in Saudi Arabia, and we propose the hypotheses below:

**Hypothesis 11:** *There is a mediating role of organizational identification in the relationship between corporate social responsibility and employee performance.*

**Hypothesis 12:** *There is a mediating role of organizational identification in the relationship between corporate social responsibility and turnover intention.*

As a consequence of this, the conceptual framework that can be seen in Figure 1 was built based on the literature study that was shown earlier.

Corporate social responsibility (CSR) refers to a company's responsibility to operate ethically and sustainably that considers the impact of its actions on society and the environment [85].

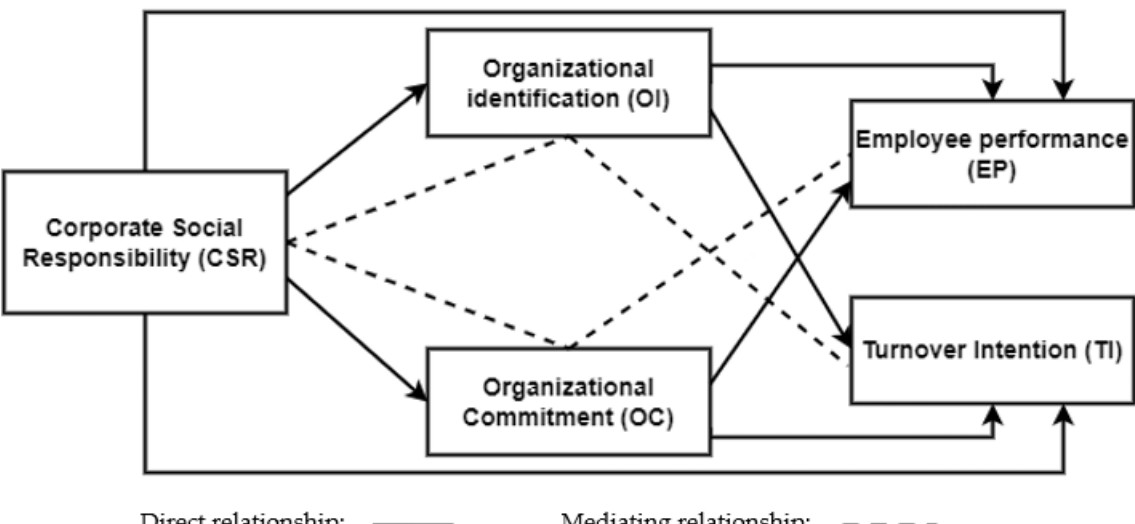

**Figure 1.** Research Framework.

Organizational identification (OI) refers to how an individual identifies with and feels a sense of belonging to an organization. It is a psychological process involving shared values, goals, and identity between individuals and organizations [73].

Organizational commitment (OC) refers to an individual's psychological attachment to an organization. It is a multidimensional construct with economic, affective, and moral aspects [86].

Employee performance (EP) refers to the duties and tasks workers are paid to complete. It includes actions that fulfill the job description's requirements [18].

Turnover intention (TI) describes a worker's plans to quit their present employer [72].

*2.10. Theoretical Underpinning*

The social identity theory was founded in 1979 to develop a psychological understanding of intergroup discrimination. [87]. As was mentioned earlier, social identity theory (SIT) is widely used to understand and predict the relationship between corporate social responsibility (CSR) and its related outcomes, such as organizational identification (OI) and organizational commitment. This is because SIT assumes individuals have a shared sense of who they are and where they come from (OC) [41,88].

SIT is a theoretical framework that explains how individuals identify with groups and how this identification influences their attitudes and behaviors toward the group. In the context of corporate social responsibility (CSR), social identity theory proposes that employees identify with their organization and that this identification is influenced by the CSR initiatives that their organization engages in. In other words, employees see their jobs as contributing to the greater good. The relationship between CSR and employee outcomes (turnover intention, employee performance) is mediated by organizational identification and organizational commitment, which refers to the degree to which employees feel a sense of belonging and attachment to their organization. The relationship between CSR and employee outcomes (turnover intention, employee performance) is shown in Figure 1 [89–91]. One of the most important mediators between CSR and employee performance and intention to leave their current position is organizational identification and organizational commitment. Employees who perceive their company's CSR efforts well are more likely to identify with their employer. This, in turn, leads to beneficial outcomes such as decreased desire to leave the company, increased performance, and increased willingness to assist others.

According to social identity theory, employees identify with their organizations, and this identification affects the employees' attitudes and behaviors toward the organizations in which they work. According to the moderated mediation model, employees with high levels of moral identity are more likely to identify with their organization and ex-

perience positive outcomes (such as lower turnover intention, higher levels of in-role job performance, and increased levels of helping behavior) when they perceive high levels of corporate social responsibility (CSR) [81,90].

For further clarification, social identity theory can be applied to the context and variables of this study in the following ways: First, organizational identification: Social Identity Theory posits that individuals derive a portion of their self-concept from the groups to which they belong, including their organization. In this study, organizational identification is a critical variable. Employees who strongly identify with their organization are more likely to perceive themselves as integral members of that group. This identification can influence their attitudes and behaviors, including their performance and intentions related to turnover. Second, organizational commitment: social identity theory suggests that when individuals identify strongly with an organization, they are more likely to be committed to its goals and values. In this study, organizational commitment is another central variable. Employees who identify with their organization and its CSR initiatives may display higher levels of commitment, which, in turn, can affect their performance positively and reduce their intentions of leaving the organization.

In general, social identity theory offers a helpful framework for comprehending the connection between CSR and employee outcomes, drawing attention to the significance of organizational identification and commitment in mediating this connection.

## 3. Method

### 3.1. Design and Procedures

This study uses a quantitative cross-sectional research design. As Bryman and Bell [92] outlined, quantitative research steps may be a linear progression from theory to findings. The study determined the suitable sample size using Krejcie and Morgan [93] table for sample size determination. Given that the population of employees within the Saudi Arabian banking sector was reported as 47,181 by the Saudi Arabian Monetary Authority (SAMA) in 2020, the minimum required sample size was calculated to be 381.

There, 550 employees participated in this questionnaire, representing all of the listed banks in the Riyadh area of Saudi Arabia. Riyadh was chosen since many financial institutions' offices are located there. While visiting the banks, respondents filled out the survey questionnaires. Between March 2020 and February 2021, the survey was made available offline and online.

Questions on the participants' demographics, level of interest in corporate social responsibility (CSR), organizational identification (OI), organizational commitment (OC), employee performance (EP), and desire to leave the company turnover intention (TI) were all addressed. The research found that the common method variance would increase if the same people were asked about all the variables without sectioning, undermining the study's internal validity [94]. We took several steps to minimize common method variance in our study. First, we designed separate questionnaire sections for each variable, ensuring participants were not asked about all variables in a single section. Secondly, to provide context and emphasize the relevance of questions, we included clear introductory statements at the beginning of each section.

Additionally, we assured respondents of the anonymity and confidentiality of their responses to minimize social desirability bias. Finally, we employed statistical techniques like factor analysis and covariance matrix examination to detect and control for potential methodological bias in our data analysis. These measures collectively created a conceptual divide between independent, mediator, and dependent variables, reducing common method variance and enhancing the internal validity of our study.

### 3.2. Scales

This section had three demographic questions: sexual orientation, age, and education degree. This section was designated at the outset of the survey's creation process. The four components were measured in this research using validated and reliable instruments (CSR,

OI, OC, TI, and EP). These standards were revised and customized to meet the needs of this study. Every metric was evaluated on a 5-point Likert scale (1 = strongly disagree, 2 = disagree, 3 = neutral, 4 = agree, 5 = strongly agree). The current research used Arabic and English versions of the questionnaire.

### 3.2.1. Corporate Social Responsibility

Standardized measures, such as Turker's 17-item scale, were used to evaluate CSR [48]. Many CSR studies have utilized this scale [41,95,96]. The response scale is a 1–5 Likert scale.

### 3.2.2. Organizational Identification

The scale developed by Mael and Ashforth Mael and Ashforth [97] was used to measure OI. The six-item scale has seen extensive application in OI research—for example Ates et al., in [98].

### 3.2.3. Organizational Commitment

A condensed version with just nine items was used to test OC. The nine-item shorter version of the OCQ was established by Mowday Mowday [99], and it is regarded as one of the most widely used and trustworthy measures of OC.

### 3.2.4. Employee Performance

Koopmans et al. Koopmans [100] developed a scale to assess EP. Throughout the preceding literature analysis, four things were used rather often. The concept, directly and indirectly, assesses employee performance by mediating the links between organizational commitment, identity, and business reputation.

### 3.2.5. Turnover Intention

The TI 3-items are an adaptation of Jacobs and Roodt Jacobs and Roodt [101] questions, and they use a Likert scale with five points that range from "1" (strongly disagree) to "5" (strongly agree).

As a consequence of this, the research project consisted of three stages of validating questionnaire surveys. The first step is the validity of the survey in its appearance and content. The pilot study, which is a good technique for reviewing a measuring instrument in order to enhance the reliability and validity of the tools before the major research has been carried out, is the focus of the second part of the research project [102]—lastly, the primary survey is conducted to test the theoretical hypothesis.

In this present investigation, 550 bank employees from Saudi Arabia took part voluntarily, providing informed consent. The respondents' demographic distribution is 86.6% male and 13.4% female. The age group between 26 and 30 constituted the largest proportion, accounting for 28.7% of the participants. Regarding education, the majority held a bachelor's degree, comprising 61.5% of the respondents. While response rates for online and self-administered questionnaires have decreased over time [103], the researcher achieved a high response rate with the targeted sample in this study. Ethically informed consent was obtained from all subjects involved in the study. They voluntarily agreed to participate in this study, and we guaranteed them the data's confidentiality.

## 4. Results

### 4.1. Common Method Bise

If the same respondents were used for each variable, the common method variance may increase, which is detrimental to the study's internal validity [94]. Consequently, a psychological distinction has been created between the study's independent factors, mediator variables, and dependent variables to lessen the possibility of variation caused by a shared approach. Therefore, to avoid giving the impression that the variables were interdependent on one another, this research presented the numerous variables in distinct portions and assessed for common method bias. A measurement inaccuracy that

may impact the study's reliability is referred to as common method bias. It represents the systematic variation in error associated with estimated and measured variables [104]. Table 1 displays the distribution of variances for the variables under consideration. When we examine these variables collectively, they account for only 37.649 percent of the total variation. This percentage falls below the commonly accepted threshold of 50% [104]. This outcome implies that these results are less likely to have been significantly affected by common method bias, which is typically a concern when a single method or source dominates the data collection process. In our case, since the explained variance is below 50%, it suggests that common method bias is less likely to be a major influence on our results.

**Table 1.** Total variance explained.

| Extraction Sums of Squared Loadings | | |
|---|---|---|
| **Total** | **% of Variance** | **Cumulative %** |
| 18.072 | 37.649 | 37.649 |

*4.2. Measurement Model*

4.2.1. Convergent Validity

In order to evaluate the measured model, both reliability and validity tests were carried out. During the reliability test, four different indicators were evaluated. These were outer loadings (OL), Cronbach's alpha (CA), composite reliability (CR), and average variance extracted (AVE). For each item, the values of CA [105], CR [106], and OL [107] must be more than 0.70. AVE value, and the AVE needs to be more than 0.5 for every build [108,109].

Table 2 and Figure 2 show that the OL value was between 0.727 and 0.960 (above 0.700), except for the items CSR3 and CSR6, which were eliminated due to their outer loading being lower than 0.70. The value of CA ranged from 0.802 to 0.962, which indicates that it was higher than the benchmark value of 0.70. Both CR (rho a) and CR (rho c) values were more than 0.700 and varied from 0.802 to 0.966. In conclusion, the value of AVE ranged from 0.655 up to 0.866, which is higher than the standard of 0.500.

**Table 2.** Reliability and validity of the model.

| Constructs | OL | Cronbach's Alpha | CR (rho_a) | CR (rho_c) | (AVE) |
|---|---|---|---|---|---|
| **CSR** | | 0.962 | 0.963 | 0.966 | 0.655 |
| CSR.1 | 0.734 | | | | |
| CSR.10 | 0.850 | | | | |
| CSR.11 | 0.790 | | | | |
| CSR.12 | 0.758 | | | | |
| CSR.13 | 0.776 | | | | |
| CSR.14 | 0.793 | | | | |
| CSR.15 | 0.819 | | | | |
| CSR.16 | 0.810 | | | | |
| CSR.17 | 0.887 | | | | |
| CSR.2 | 0.773 | | | | |
| CSR.4 | 0.751 | | | | |
| CSR.5 | 0.877 | | | | |
| CSR.7 | 0.795 | | | | |
| CSR.8 | 0.810 | | | | |
| CSR.9 | 0.890 | | | | |
| CSR.3 | Delete | | | | |
| CSR.6 | Delete | | | | |

**Table 2.** *Cont.*

| Constructs | OL | Cronbach's Alpha | CR (rho_a) | CR (rho_c) | (AVE) |
|---|---|---|---|---|---|
| **EP** | | 0.802 | 0.802 | 0.871 | 0.627 |
| EP.1 | 0.749 | | | | |
| EP.2 | 0.824 | | | | |
| EP.3 | 0.790 | | | | |
| EP.4 | 0.804 | | | | |
| **OC** | | 0.943 | 0.946 | 0.952 | 0.686 |
| OC.1 | 0.727 | | | | |
| OC.2 | 0.837 | | | | |
| OC.3 | 0.823 | | | | |
| OC.4 | 0.820 | | | | |
| OC.5 | 0.839 | | | | |
| OC.6 | 0.836 | | | | |
| OC.7 | 0.875 | | | | |
| OC.8 | 0.834 | | | | |
| OC.9 | 0.856 | | | | |
| **OI** | | 0.912 | 0.924 | 0.932 | 0.698 |
| OI.1 | 0.759 | | | | |
| OI.2 | 0.861 | | | | |
| OI.3 | 0.899 | | | | |
| OI.4 | 0.892 | | | | |
| OI.5 | 0.733 | | | | |
| OI.6 | 0.851 | | | | |
| **TI** | | 0.922 | 0.929 | 0.951 | 0.866 |
| TI.5 | 0.960 | | | | |
| TI.6 | 0.882 | | | | |
| TI.7 | 0.947 | | | | |

OL: Outer loadings, CR: composite reliability, AVE: average variance extracted, CSR: corporate social responsibility, EP: employee performance, OC: organizational commitment, OI: organizational identification, TI: turnover intention.

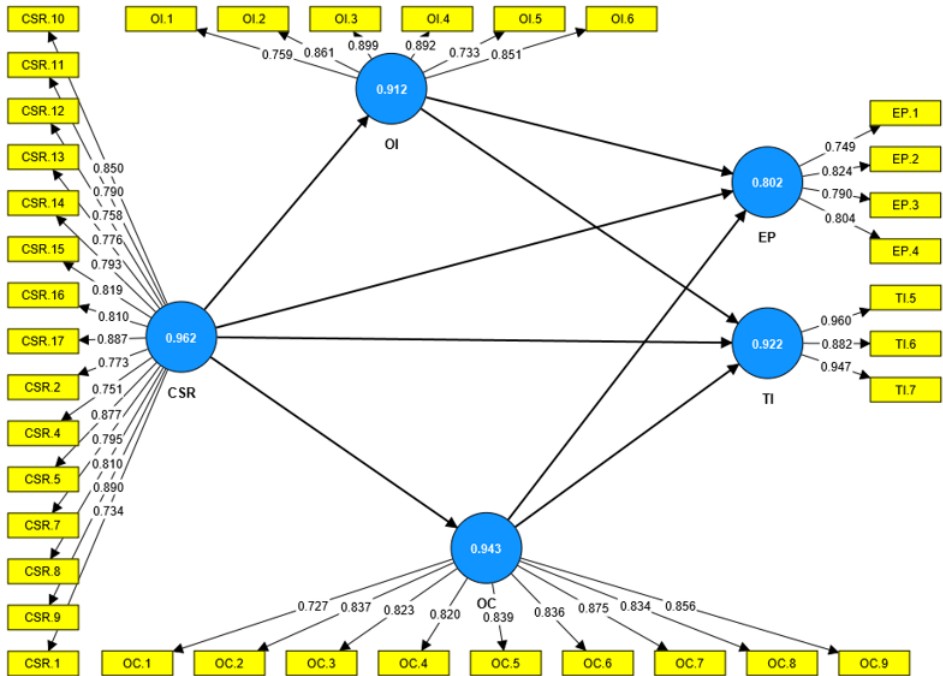

**Figure 2.** Measurement model.

4.2.2. Discriminant Validity

In the field of research, a concept known as discriminant validity is used to examine whether or not a measurement can be distinguished from other measurements with which it should not be associated. One construct validity is called content validity, and it evaluates whether or not a measure is measuring what it was designed to measure rather than something else [110].

Because it helps verify that a measure is not confused with other measures and assesses a distinct concept, discriminant validity is an essential component of any measurement system. Various methods can be used to evaluate discriminant validity, one is to examine the correlation between two measures after accounting for measurement error [111]. The heterotrait–monotrait ratio (HTMT), the Fornell–Larcker criteria, and cross-loading are the three tests this research endeavor conducts to finish the discriminant validity test.

In this study, we employed the heterotrait–monotrait ratio (HTMT) of correlations, as presented in Table 3. The HTMT is a widely recognized method for assessing discriminant validity in research [112]. Unlike the Fornell–Larcker criteria and (partial) cross-loadings, the HTMT test of correlations is often considered to provide a higher-quality evaluation of discriminant validity. To determine discriminant validity, we applied a threshold of 0.9 as this investigation's criteria, as prior research suggested [113–116]. Specifically, we examined the HTMT ratios to assess whether the correlations between different constructs were below this threshold, indicating that they were sufficiently distinct from each other.

**Table 3.** Heterotrait–monotrait ratio (HTMT) matrix.

|  | **CSR** | **EP** | **OC** | **OI** | **TI** |
|---|---|---|---|---|---|
| CSR |  |  |  |  |  |
| EP | 0.509 |  |  |  |  |
| OC | 0.680 | 0.672 |  |  |  |
| OI | 0.426 | 0.702 | 0.765 |  |  |
| TI | 0.449 | 0.216 | 0.541 | 0.323 |  |

We also considered the Fornell–Larcker criteria, which state that the average variance extracted (AVE) square root for a specific construct should be greater than its correlation with all other constructs [115,117]. This comparison, found in Table 4, provides an additional layer of validation for the distinctiveness of the constructs under study.

**Table 4.** Fornell–Larcker criterion.

|  | **CSR** | **EP** | **OC** | **OI** | **TI** |
|---|---|---|---|---|---|
| CSR | 0.809 |  |  |  |  |
| EP | 0.450 | 0.792 |  |  |  |
| OC | 0.657 | 0.586 | 0.828 |  |  |
| OI | 0.408 | 0.602 | 0.707 | 0.835 |  |
| TI | −0.427 | −0.188 | −0.511 | −0.306 | 0.931 |

Finally, the cross-loading test was performed, and the results may be seen in Table 5. The cross-loading criteria are the third approach used in this investigation, and it was also utilized to evaluate the discriminant validity of the variables. This technique establishes whether indicators loading on a particular latent construct should be greater than on any other constructs per line [118]. This indicates that the loading of indications or objects onto the primary construct must be bigger than any other construction. According to the findings shown in Table 5, the loading of all latent variables (indicators) is superior to that of other constructs when compared row by row. The findings demonstrate a significant one-dimensionality for each examined component [119]. Overall, this comprehensive approach enhances our confidence that the constructs in our study are distinct and not unduly influenced by shared variance.

**Table 5.** Cross loadings.

|        | CSR    | EP     | OC     | OI     | TI      |
| ------ | ------ | ------ | ------ | ------ | ------- |
| CSR.1  | **0.734** | 0.324  | 0.471  | 0.281  | −0.269  |
| CSR.10 | **0.850** | 0.335  | 0.563  | 0.354  | −0.371  |
| CSR.11 | **0.790** | 0.364  | 0.537  | 0.353  | −0.310  |
| CSR.12 | **0.758** | 0.386  | 0.485  | 0.315  | −0.323  |
| CSR.13 | **0.776** | 0.338  | 0.470  | 0.270  | −0.280  |
| CSR.14 | **0.793** | 0.375  | 0.521  | 0.314  | −0.367  |
| CSR.15 | **0.819** | 0.302  | 0.561  | 0.360  | −0.386  |
| CSR.16 | **0.810** | 0.393  | 0.536  | 0.343  | −0.362  |
| CSR.17 | **0.887** | 0.380  | 0.561  | 0.317  | −0.385  |
| CSR.2  | **0.773** | 0.411  | 0.543  | 0.371  | −0.364  |
| CSR.4  | **0.751** | 0.338  | 0.501  | 0.312  | −0.288  |
| CSR.5  | **0.877** | 0.365  | 0.557  | 0.318  | −0.378  |
| CSR.7  | **0.795** | 0.357  | 0.484  | 0.284  | −0.284  |
| CSR.8  | **0.810** | 0.400  | 0.569  | 0.387  | −0.384  |
| CSR.9  | **0.890** | 0.384  | 0.587  | 0.352  | −0.388  |
| EP.1   | 0.422  | **0.749** | 0.517  | 0.469  | −0.177  |
| EP.2   | 0.319  | **0.824** | 0.428  | 0.482  | −0.130  |
| EP.3   | 0.329  | **0.790** | 0.408  | 0.442  | −0.149  |
| EP.4   | 0.348  | **0.804** | 0.491  | 0.509  | −0.137  |
| OC.1   | 0.370  | 0.494  | **0.727** | 0.666  | −0.265  |
| OC.2   | 0.512  | 0.515  | **0.837** | 0.679  | −0.419  |
| OC.3   | 0.595  | 0.487  | **0.823** | 0.478  | −0.434  |
| OC.4   | 0.536  | 0.502  | **0.820** | 0.610  | −0.426  |
| OC.5   | 0.466  | 0.502  | **0.839** | 0.681  | −0.397  |
| OC.6   | 0.699  | 0.464  | **0.836** | 0.490  | −0.463  |
| OC.7   | 0.605  | 0.448  | **0.875** | 0.556  | −0.514  |
| OC.8   | 0.475  | 0.493  | **0.834** | 0.636  | −0.389  |
| OC.9   | 0.578  | 0.482  | **0.856** | 0.536  | −0.455  |
| OI.1   | 0.282  | 0.485  | 0.503  | **0.759** | −0.207  |
| OI.2   | 0.345  | 0.514  | 0.594  | **0.861** | −0.275  |
| OI.3   | 0.408  | 0.545  | 0.675  | **0.899** | −0.314  |
| OI.4   | 0.382  | 0.523  | 0.668  | **0.892** | −0.284  |
| OI.5   | 0.201  | 0.441  | 0.460  | **0.733** | −0.112  |
| OI.6   | 0.386  | 0.506  | 0.605  | **0.851** | −0.298  |
| TI.5   | −0.444 | −0.214 | −0.512 | −0.320 | **0.960** |
| TI.6   | −0.370 | −0.145 | −0.443 | −0.234 | **0.882** |
| TI.7   | −0.372 | −0.161 | −0.467 | −0.297 | **0.947** |

*4.3. Structural Model Assessment*

4.3.1. The Strength of the Structural Model as an Explanatory Tool

In our research, $R^2$ values are used to assess the proportion of variance in the endogenous variables (dependent variables) that our independent variables or predictors can explain. These values indicate our model's strength and effectiveness in explaining the observed outcomes [120]. The $R^2$ measures the model's explanatory power [121], also known as in-sample predictive power. It is a representation of the variation that is explained in each of the endogenous components [122]. The $R^2$ may take on any value between 0 and 1, with higher values suggesting a more substantial capacity for explanation. According to a general rule of thumb that was proposed by Cohen Cohen [123], the $R^2$ values for endogenous latent variables should be evaluated as follows: 0.26 (substantial), 0.13 (moderate), and 0.02 (weak) [124]. The findings are shown in Table 6, and it can be seen that the R2 value for each of the endogenous constructs is more than 0.26. This indicates that the model's capacity for explanation is significant [123]. Significant $R^2$ values indicate that the study model effectively explains variations in the endogenous variables, providing insights into the factors that influence these outcomes. The practical significance lies in the implications for decision-making and strategy development, as it highlights

which variables have a substantial impact and should be prioritized for intervention or improvement efforts in the relevant context.

**Table 6.** Explanatory power.

| Predictor | Outcomes | R2 | F2 | Q2 |
|:---:|:---:|:---:|:---:|:---:|
| CSR | ≥EP | | 0.023 | |
| OC | ≥EP | 0.427 | 0.025 | 0.199 |
| OI | ≥EP | | 0.134 | |
| CSR | ≥OC | 0.432 | 0.761 | 0.430 |
| CSR | ≥OI | 0.167 | 0.200 | 0.163 |
| CSR | ≥TI | | 0.018 | |
| OC | ≥TI | 0.280 | 0.106 | 0.179 |
| OI | ≥TI | | 0.006 | |

Effect sizes, specifically f-squared ($f^2$), provide valuable insights into the magnitude of the relationships or differences observed in a study [125]. The magnitude of the impact exerted by each independent variable on the study's outcome is referred to as the effect size. When an independent variable is removed from the PLS path model, the model then calculates the variance in squared correlation values to determine whether or not the removed independent variable had a significant impact on the value of the dependent variable. If f-sq is 0.35, then the influence of the predictor variable is large at the structural level. If f-sq 2 is 0.15, then the impact of the predictor variable is medium, and if f-sq is 0.02, then the effect of the predictor variable is minor [123]. To put it another way, the effect size is an evaluation of the amount or strength of the link between the latent variables. According to the findings shown in Table 6, the f-square impact size varied widely from 0.006 (considered minimal) for OI on TI to 0.761 (considered substantial) for CSR on OC, which refers to the size or strength of the relationship between these variables within the context of this study. When we have substantial effect sizes, it means the relationships or differences identified in this study are not only statistically significant but also hold practical significance in the context of this research. These findings can have real-world implications, influencing decision-making and strategies within organizations or relevant fields.

In conclusion, the Q2 values for the endogenous constructs were greater than 0, and as a result, predictive significance was determined. If your model contains Q-square values greater than zero, this indicates that the model has predictive significance and that your values have been properly rebuilt. When Q2 is greater than 0, it suggests that the model has some predictive value. Explanatory power for the structural model, which comprises the R2, F2, and Q2, is shown in Table 6.

### 4.3.2. Direct Effect

In the process of testing hypotheses, a total of twelve hypotheses were examined; of these, eight were examined for direct correlations, while the other four were examined for mediating interactions. CSR was the independent factor, TI and EP were the dependent variables in the structural model framework, and OI and OC were the mediating variables. A direct link existed between these four sets of variables.

The 5000 bootstrap samples selected for "Sign No Changes" and "Complete Bootstrapping" are put through their paces. The bias acceleration and correction process was used in the last step of the bootstrapping operation. Additionally, the advanced settings carried out a one-tailed significance level test at the 0.05 probability level. It can be seen from Table 7 and Figure 3 that eight of the hypotheses are supported.

**Table 7.** Direct effect summary.

| Hypotheses | Path Relationships | Original Sample (O) | STDEV | T Value | *p* Values | Result |
|---|---|---|---|---|---|---|
| H1 | CSR ≥ EP | 0.154 | 0.046 | 3.374 | 0.001 | Supported |
| H2 | CSR ≥ OC | 0.657 | 0.024 | 27.916 | 0.000 | Supported |
| H3 | CSR ≥ OI | 0.408 | 0.034 | 11.867 | 0.000 | Supported |
| H4 | CSR ≥ TI | −0.151 | 0.048 | 3.140 | 0.002 | Supported |
| H5 | OC ≥ EP | 0.207 | 0.057 | 3.647 | 0.000 | Supported |
| H6 | OC ≥ TI | −0.476 | 0.058 | 8.243 | 0.000 | Supported |
| H7 | OI ≥ EP | 0.394 | 0.049 | 8.005 | 0.000 | Supported |
| H8 | OI ≥ TI | 0.092 | 0.045 | 2.054 | 0.040 | Supported |

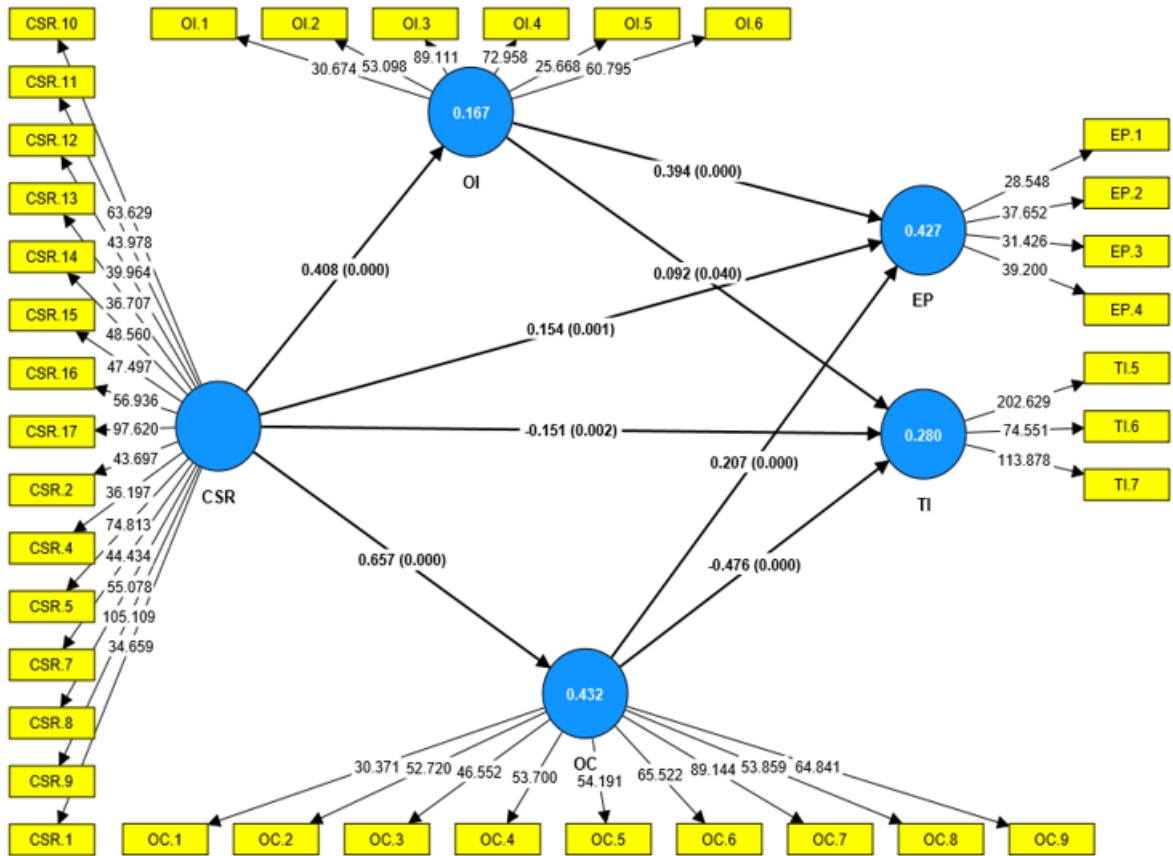

**Figure 3.** Structure model test.

### 4.3.3. Mediation Analysis

The current investigation used bootstrapping to substantiate the mediating impact [126]. Several academics have advised this methodology for studies examining the impact of such indirect impacts [127]. In addition, it is believed that bootstrap findings have more accurate probability estimates. This is because this approach helps overcome mediation difficulties and does not need a normal distribution of the mediator and outcome variables [128]. As a result, this strategy is validated for two different reasons. The first advantage is that it effectively evaluates the significance and confidence intervals in various contexts. Another advantage of using this strategy is that it does not need to create many assumptions. Therefore, the findings achieved through this approach better reflect the true situation [129]. The results of the indirect impact are shown in Table 8, and it can be seen that all of the mediating hypotheses were validated.

**Table 8.** Analysis of mediation.

| H | Relationship | Indirect Effect | | *p* Values | Confidence Interval | | Decision |
|---|---|---|---|---|---|---|---|
| | | Original Sample | T Statistics | | LL 95% | UL 95% | |
| H9 | CSR ≥ OC ≥ EP | 0.136 | 3.637 | 0.000 | 0.077 | 0.199 | Mediated |
| H10 | CSR ≥ OC ≥ TI | −0.313 | 7.360 | 0.000 | −0.384 | −0.247 | Mediated |
| H11 | CSR ≥ OI ≥ EP | 0.161 | 6.558 | 0.000 | 0.122 | 0.203 | Mediated |
| H12 | CSR ≥ OI ≥ TI | 0.038 | 1.991 | 0.023 | 0.008 | 0.070 | Mediated |

Our validated mediation findings have significant implications in the context of the banking sector. They deepen our understanding of how corporate social responsibility (CSR) influences employee performance and turnover intention through specific pathways. For instance, the identification of organizational identification and commitment as moderators highlights their crucial roles in shaping these relationships. This has practical importance, guiding organizations in optimizing CSR strategies to enhance employee performance and reduce turnover intention. Additionally, these findings contribute to the theoretical framework of CSR by providing empirical support and insights into its impact on employee outcomes. Thus, they enrich our comprehension of this complex relationship and provide a basis for practical actions.

Consequently, our study's statistically significant findings have practical implications, including the potential to enhance employee performance and reduce turnover. CSR can be a valuable tool for organizations to boost productivity and retain talent. Additionally, we emphasize the significance of organizational identification and commitment as moderating factors, offering insights for tailoring CSR strategies to improve employee engagement. Overall, our research provides actionable guidance for strategic decision-making within organizations and underscores the practical relevance of our work.

## 5. Discussion

This research endeavor sought to unravel the multifaceted impact of corporate social responsibility (CSR) on key organizational facets, including organizational identification, organizational commitment, employee performance, and turnover intention. In the forthcoming discussion, this study scrutinizes the findings through various lenses, drawing upon previous literature and culminating in practical recommendations for organizations. By exploring these dimensions, we aim to comprehensively understand the intricate relationship between CSR, employee performance, and turnover intention.

Our findings support H1, as we observed a significant positive relationship between CSR and EP. Employees who perceived their organization as socially responsible demonstrated higher levels of job performance. This result aligns with previous research [24–26], which suggested that employees tend to be more motivated and engaged when they believe their company is committed to ethical and socially responsible practices. The positive impact of CSR on employee performance can be attributed to the social identity theory. This theory posits that employees identifying with a socially responsible organization are more likely to internalize its values and mission, leading to increased effort and dedication in their roles [25].

In H2, this study found a significant positive relationship between CSR and OI. Employee engagement in CSR initiatives has the potential to enhance employee satisfaction through these activities, consequently leading to an increase in the level of organizational identification. These results align with prior studies that have noted the importance of CSR in improving OI [29,30,35,37,38]. When employees perceive their companies as socially responsible organizations, their sense of belongingness can be improved, positively impacting their self-concepts [31].

In H3, consistent with previous research [33,41,42], our results revealed a significant correlation between CSR and OC. Existing empirical research has consistently supported a positive relationship between CSR and OC [41,43–47]. Strong and reliable employee–

employer relationships serve as the foundation for thriving corporations. Companies must cultivate an environment that attracts and retains talented and dedicated employees to achieve sustainable success in today's highly competitive business landscape.

While in H4, the research findings indicated a strong negative connection between CSR and TI, these findings were found to be negatively statistically significant. These findings align with earlier research [50–52]. If an organization's workers believe their employer prioritizes internal corporate social responsibility, they are more committed to their jobs and less inclined to seek other employment opportunities. On the other hand, if workers believe that their employer is engaging in hypocrisy, this may inversely impact their behavior and raise their desire to leave the company. After that, in H5, the current research results found a positive significant relationship between the OC and EP. These results aligned with previous research [55,57–60]. The connection between organizational commitment and job performance tends to be stronger than in individualistic cultures. Organizational commitment contributes significantly to improving job performance among male skilled professionals. Organizational commitment has also been found to considerably influence performance and moderate the association between total job stress and performance. Previous research has shown that workers' perceptions of their organizations' support are favorably associated with their organizational commitment and adversely related to their desire to leave their current positions and turnover intention [63–65]. Thus, H6 explains the observed relationship between OC and TI.

In H7, our findings revealed a significant positive relationship between OI and EP. These results are consistent with previous studies [68–70]. Employees with a strong sense of organizational identification are inclined to invest extra effort in fulfilling their roles. Their strong connection to the organization fosters a high valuation of their organizational membership. This, in turn, makes them more inclined to adhere to organizational rules and act in their best interests.

In H8, there is a relationship between organizational identification and turnover intention, as employees who identify more strongly with their organization are less likely to have turnover intentions. These results aligned with many researchers [72–74]. According to Otaye Otaye [72], organizational identification was positively related to organizational citizenship behavior, which predicts turnover intention [72].

The link between CSR, EP, and TI is mediated by organization identification in both H9 and H10 of the model. These two hypotheses are supported by the findings, which indicate that organizational identification is a partial mediator of the link between CSR and employee performance as well as turnover intention. This research suggests that when workers consider their firm socially responsible, they are more likely to identify with the organization. This, in turn, improves their job performance and reduces the likelihood that they would want to quit the organization. This study is consistent with social identity theory, which suggests that organizational identification is a psychological mechanism that ties CSR attitudes to employee performance [130].

As implications of mediation, H9 suggests that when employees perceive their organization as socially responsible, they are more likely to connect with it emotionally, resulting in improved job performance. This emotional attachment motivates employees to excel in their roles and reduces the likelihood of them considering leaving the organization. This finding aligns well with social identity theory (SIT), which posits that employees derive a significant part of their identity from affiliation with a particular organization. Similarly, Hypothesis 10 (H10) reveals that when employees view their organization as socially responsible, they tend to identify with it more strongly, fostering a sense of belonging and attachment. This heightened sense of affiliation translates into reduced intentions to quit, as employees who identify with the organization are less inclined to seek employment elsewhere.

Similarly, the association between CSR, EP, and TI is mediated by organizational commitment in H11 and H12, respectively. According to our findings, organizational commitment does play a major role in the process of partial mediation. According to

this result, an employee's impression of corporate social responsibility (CSR) positively impacts their commitment to the firm, affecting their work performance and desire to quit the organization. This conclusion is compatible with the SIT theory, which is generally acknowledged as the fundamental theory used to grasp and foresee the link between CSR and its related effects, such as OC. Specifically, this result is consistent with the SIT theory in the following way [41,48]. According to the theory, an individual or group's perceptions of their identity and beliefs regarding their organization can strengthen their identification with that organization. This holds whether the individual or group is the one doing the perceiving or the group is the one doing the believing [131].

As implications of mediation, in this scenario of (H11), CSR activities positively influence an employee's commitment to the organization. This heightened commitment, in turn, leads to enhanced job performance. Committed employees are more likely to go the extra mile to contribute to their organization's success, and when CSR initiatives align with their values and beliefs, it reinforces their commitment. Lastly, in (H12), when employees perceive CSR efforts positively, it strengthens their commitment to the organization, reducing their intention to leave. This finding underscores the notion that organizational commitment is a strong bond that ties employees to their workplace. When CSR activities resonate with employees, they reinforce their commitment and discourage thoughts of leaving the organization.

In conclusion, this study offers valuable insights into the significant influence of corporate social responsibility (CSR) on various aspects of employee engagement and retention within organizations. Our findings underscore several key takeaways: firstly, CSR positively impacts employee performance (EP), as employees who perceive their organization as socially responsible tend to exhibit higher job performance. Secondly, CSR fosters organizational identification (OI) by enhancing employee satisfaction and a sense of belonging to the organization. Thirdly, CSR bolsters organizational commitment (OC), contributing to the foundation of thriving organizations. Lastly, CSR reduces turnover intention (TI), encouraging employee commitment and reducing the desire to seek alternative employment.

To translate these findings into practical recommendations for organizations, it is crucial to prioritize CSR initiatives that align with the organization's values and mission. Actively involving employees in CSR activities and emphasizing their engagement can strengthen their sense of identification with the organization. Promoting a culture of commitment, valuing employee contributions, and ensuring CSR efforts resonate with employees can enhance organizational commitment. Additionally, organizations should maintain transparency and consistency in CSR efforts to address perceptions of hypocrisy that can negatively impact employee attitudes. Regular measurement and monitoring of the impact of CSR initiatives on employee performance, identification, and commitment are essential for refining and improving CSR strategies. By implementing these recommendations, organizations can enhance their CSR practices and cultivate a more engaged and committed workforce, ultimately leading to improved organizational performance and reduced turnover rates.

## 6. Conclusions

In conclusion, this research aimed to assess the roles of organizational identification and commitment as mediators in the relationship between corporate social responsibility and employee performance, as well as employees' intention to leave their current positions (turnover intention). The study tested 12 hypotheses, with 8 involving direct relationships and 4 exploring mediated relationships. The findings indicate that all hypotheses were thoroughly examined and evaluated. As a result of this empirical research conducted within the banking industry in Saudi Arabia, the study yields significant implications, both theoretical and practical, based on the precise results obtained.

### 6.1. Theoretical Contributions

The study makes a significant theoretical contribution to the existing literature by examining the roles of organizational identification and commitment as mediators in the relationship between corporate social responsibility and employee performance, as well as turnover intention among employees in the Saudi Arabian banking sector. Furthermore, this research highlights the crucial significance of organizational identification and commitment as essential mediators in linking corporate social responsibility initiatives to employee performance and their inclination to stay with the organization. Overall, this study enhances our theoretical understanding of the factors influencing employee performance and turnover intention within the context of the banking industry in Saudi Arabia.

### 6.2. Practical Implications

The research findings can serve as a valuable resource for firms within the banking industry in Saudi Arabia, aiding them in creating and implementing corporate social responsibility programs. These initiatives have the potential to boost employee performance while reducing employees' intentions to leave their positions. The research underscores the paramount importance of fostering a sense of organizational identification and commitment among staff members. Furthermore, these findings can be leveraged in the development of tailored training programs for employees in the Saudi Arabian banking sector. These programs should strongly emphasize enhancing employees' identification with the organization and their commitment to its mission and values.

### 6.3. Limitations of Study

Despite this study's numerous contributions, future researchers must consider several limitations. The research design employed here is cross-sectional, meaning it collects data at a specific moment. This design hinders the establishment of causal linkages between variables, potentially missing relationship changes over time. Consequently, longitudinal studies are better suited to capture these temporal changes, providing a deeper and more comprehensive understanding of the linkages.

This study primarily focuses on the Saudi Arabian banking industry, which limits the generalizability of its results to other industries or nations. The conclusions drawn are particularly relevant within Saudi Arabia and the banking sector. To assess the results' applicability to different cultural settings or fields and to enhance external validity, researchers may explore cross-cultural studies that compare findings with similar industries in other countries. Collaboration with experts in those regions or industries can offer valuable insights for tailoring recommendations to specific needs.

While this research investigates organizational identification and commitment as potential mediators, it is crucial to consider other potential mediators or moderators that may influence the relationship between corporate social responsibility and employee performance, as well as turnover intention. Variables such as leadership styles, organizational culture, and job satisfaction should also be taken into account. In addition, future research could potentially explore the legislative aspects of the manager-employer relationship and labor law issues, as these are important factors that many studies recommend addressing [132–134].

**Author Contributions:** Conceptualization, M.A.; methodology, M.A.; software, M.A. and A.-B.A.A.-M.; validation, M.A. and A.-B.A.A.-M.; formal analysis, M.A. and A.-B.A.A.-M.; investigation, M.A.; data curation, M.A.; writing—original draft preparation, M.A. and A.-B.A.A.-M.; writing—review and editing, M.A. and A.-B.A.A.-M.; project administration, M.A.; funding acquisition, M.A. All authors have read and agreed to the published version of the manuscript.

**Funding:** This work was supported and funded by the Deanship of Scientific Research at Imam Mohammad Ibn Saud Islamic University (IMSIU) (grant number IMSIU-RG23106).

**Institutional Review Board Statement:** Not applicable.

**Informed Consent Statement:** Informed consent was obtained from all subjects involved in the study.

**Data Availability Statement:** Data available upon reasonable request.

**Acknowledgments:** Authors would like to thank Imam Mohammad Ibn Saud Islamic University for supporting this study. Also many thanks for those who participated in this study from banking sector.

**Conflicts of Interest:** The authors declare no conflict of interest.

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
