# Peer review of "The Association between Corporate Social Responsibility, Employee Performance, and Turnover Intention Moderated by Organizational Identification and Commitment"

_sustainability, doi:10.3390/su151914202_

Round 1

Reviewer 1 Report

the main weaknesses of the paper are:

w1: there are many hypothesis in this paper, it is better if the authors can provide some references of such hypothesis.

w2: table 8 has some layout problems.

w3: the analysis of this paper uses 550 surveys of employees, the method of the 550 selection is not introduced.

w4: the authors can reference techniques like LedgerDB in VLDB2020 to ensure the survey data is not tampered with, to make the result more valuable.

ok

Author Response

Response to Reviewer 1.

The Association Between Corporate Social Responsibility, Employee Performance, and Turnover Intention Moderated by Organizational Identification and Commitment

Manuscript ID:  sustainability-2613930

----------------------------------------------------------------------------------------------------------------------------------

We are very thankful for your positive reviews, comments, suggestions, and time spent reviewing this article (sustainability-2613930). Below we discuss all issues raised by you. Below, we list the reviewer’s comments, followed by our responses. The changes are highlighted in the red color in the revised manuscript:

Response to Comments from Reviewer 1:

No.

Comment

Response / Correction made

Location

1.

wthere are many hypothesis in this paper, it is better if the authors can provide some references of such hypothesis.

Thank you for your feedback.

Hypotheses are an integral part of research because they provide clear objectives, testable predictions, and scientific rigor to the study. They guide the research process, ensure focus, and help structure data collection and analysis. Moreover, hypotheses enhance the communication of research questions and findings, facilitating transparency and reproducibility. Thus, we have come up with many hypotheses from the literature review, and then we tested them by SEM to achieve the study's objectives. Moreover, all references before each hypothesis are considered as the base of it.

2.

w2: table 8 has some layout problems.

Thank you for your important comment.

The layout of Table 8 has been modified.

Table 8

3.

w3: the analysis of this paper uses 550 surveys of employees, the method of the 550 selection is not introduced.

Thank you for your comment.

The method of selection of the sample size has been added.

The study determined the suitable sample size using Krejcie and Morgan [92] table for sample size determination. Given that the population of employees within the Saudi Arabian banking sector was reported as 47,181 by the Saudi Arabian Monetary Authority (SAMA) in 2020, the minimum required sample size was calculated to be 381

3.1. Design and Procedures

4.

w4: the authors can reference techniques like LedgerDB in VLDB2020 to ensure the survey data is not tampered with, to make the result more valuable.

Thank you for your valuable feedback and suggestion to reference techniques like LedgerDB from VLDB2020. While we appreciate the recommendation, our research focuses on a different and objective aspect that doesn't directly involve data tampering concerns. The main concern in our data is the bise, so we run the Common method variance to ensure there is no bise in our data. Thus, when we examined these variables collectively, they accounted for only 37.649 percent of the total variation. This percentage falls below the commonly accepted threshold of 50% [103].

4.1. Common method bise

Reviewer 2 Report

1. A number of studies have validated the relationships of the variables in the model, what are the differences between this study and others?

2. CSR is an organizational level variable, while the other variables in the model are individual level variables, which is more suitable for cross-level research.

3. There is a lack of recent research in the literature part. The explanation of the mediating role is not clear enough.

4. The Method section needs to describe more detalis how to collect data in different stages. 

5. Statistical tables should be  more standardized.

6. In the Conclusion section, it is recommended to write theoretical and practical implications instead of just repeating the findings of the study.

7. Lack of discussion of research limitations.

Author Response

Response to Reviewer 2.

The Association Between Corporate Social Responsibility, Employee Performance, and Turnover Intention Moderated by Organizational Identification and Commitment

Manuscript ID:  sustainability-2613930

----------------------------------------------------------------------------------------------------------------------------------

We are very thankful for your positive reviews, comments, suggestions, and time spent reviewing this article (sustainability-2613930). Below we discuss all issues raised by you. Below, we list the reviewer’s comments, followed by our responses. The changes are highlighted in the red color in the revised manuscript:

Response to Comments from Reviewer 2:

No.

Comment

Response / Correction made

Location

1.

A number of studies have validated the relationships of the variables in the model, what are the differences between this study and others?

Thank you for your important comment. Let me justify it. Based on the literature review, there is a lack of research examining the role of organizational identification and commitment as mediators of the relationship between corporate social responsibility, employee performance, and turnover intention. Due to the scarcity of research exploring the mediating roles of OI and OC in these relationships, this study proposes that employing OC and OI as mediators will facilitate a deeper understanding of the link between the independent variable (CSR) and the dependent variables (EP and TI) in this specific context. The mechanisms through which OC and OI impact TI and EP have largely remained uncharted territory, especially within the Saudi Arabian banking sector, and have not garnered sufficient attention. Consequently, an urgent imperative exists to broaden the theoretical and empirical underpinnings to grasp the essence of the relationship between CSR, EP, TI, and the mediating factors of OI and OC.

2.

CSR is an organizational level variable, while the other variables in the model are individual level variables, which is more suitable for cross-level research.

Thank you for your feedback. However, In our study, we specifically chose to measure CSR from the employee perspective because it aligns with the primary focus of our research, which centers on understanding how CSR practices impact individual employees within an organization. By capturing CSR through the lens of individual employees, we aim to explore the direct effects of CSR on their attitudes and behaviors. Additionally, we acknowledge that CSR is typically considered an organizational-level variable. However, in the context of our research objectives, we have chosen to examine it at the individual level because it allows us to delve into the immediate impact of CSR on employee outcomes, such as job satisfaction, organizational commitment, and turnover intention. We believe this individual-level approach is more suitable for addressing our research questions, given our emphasis on examining the effects of CSR practices on individual employees.

3.

There is a lack of recent research in the literature part. The explanation of the mediating role is not clear enough.

Thank you for your important comment—

More explanation about the mediating has been added. Also, most of the literature review references were cited from the last ten years.

Literature review.

2.9. The mediating role of the OC and OI in the relationship between the CSR and EP, TI

4.

The Method section needs to describe more detalis how to collect data in different stages. 

Thank you for your comment.

The method section clearly described the developed survey and data collection stage.

3.1. Design and Procedures

3.2. Scales

Refer to lines 467-472

5

Statistical tables should be  more standardized.

We improved the standardization of our statistical tables. Thank you for the valuable feedback.

Results

6

In the Conclusion section, it is recommended to write theoretical and practical implications instead of just repeating the findings of the study.

Thank you for the valuable feedback.

The theoretical and practical implications have been clearly indicated in the conclusion.

Conclusion

6.1 Theoretical contributions

6.2 Practical implications

7

Lack of discussion of research limitations.

Thank you for your comment.

The limitation section has been improved based on your suggestion.

6.3 Limitations of study

Reviewer 3 Report

The peer-reviewed scientific study is undoubtedly a very interesting scientific work, the content of which can be beneficial not only for theory but also for practice.

  Authors should also better read the instructions for authors available on the journal's website and strictly follow the thread, so that the objectives, methods, material and finally the specific results of the scientific study are clear.

A minor shortcoming of the abstract is that the authors did not list keywords alphabetically; it is also questionable why the keywords start with a capital letter?

The introduction lacks a clearly defined main goal and secondary goals. I appreciate the setting of the hypotheses, but why aren't they set in the introduction and why are there 12 of them? This number is extremely high. Consider reducing the number of hypotheses and/or establishing research questions with subsequent unambiguous answers in the final chapter, which would fulfill the very meaning of their scientific work.

The relatively short introduction lacks a definition of the importance of the topic as well as an explanation of basic terms. Within the framework of the economy, employee performance, and research topics related to human resources, it is necessary to outline the labor law issue and devote at least one paragraph to the legislative side between the manager and the employer, the relationship of the manager himself as the representative of the employer to the employees, an explanation of the concept of who is the manager, because without this theoretical part, it will not be possible to provide a comprehensive view of the researched matter.

Thus, there is a lack of definition of these issues from a legislative point of view, as they do:

Peráček T. & Kaššaj M. (2023). The influence of jurisprudence on the formation of relations between the manager and the limited liability company. Juridical Tribune. 13 (1), pp. 43-62. 10.24818/TBJ/2023/13/1.04

Raguel, MM & Odeku, KO. 2023. Critical analysis of the failure of labor law to adequately protect atypical workers and its impact on human rights and fair labor practice. JURIDICAL TRIBUNE-TRIBUNA JURIDICA 13 (1), pp.63-81, doi: 10.24818/TBJ/2023/13/1.05

Muhamet BINAKU. 2021. VIOLATIONS OF THE RIGHTS OF EMPLOYEES IN PUBLIC AND PRIVATE SECTOR IN KOSOVO. Perspectives of Law and Public Administration, 10, (SI), pp. 133-137

To increase the scientific value of the work, I recommend the authors to expand this number by, for example, the above-mentioned works.

I also recommend supplementing the scientific study with at least one paragraph devoted to the issue of the perspective of further research on the topic in the future.

Author Response

Response to Reviewer 3.

The Association Between Corporate Social Responsibility, Employee Performance, and Turnover Intention Moderated by Organizational Identification and Commitment

Manuscript ID:  sustainability-2613930

----------------------------------------------------------------------------------------------------------------------------------

We are very thankful for your positive reviews, comments, suggestions, and time spent reviewing this article (sustainability-2613930). Below we discuss all issues raised by you. Below, we list the reviewer’s comments, followed by our responses. The changes are highlighted in the red color in the revised manuscript:

Response to Comments from Reviewer 3:

No.

Comment

Response / Correction made

Location

1.

  Authors should also better read the instructions for authors available on the journal's website and strictly follow the thread, so that the objectives, methods, material and finally the specific results of the scientific study are clear.

Thank you for your feedback.

We followed the journal's guidelines, and the manuscript improved accordingly.

Whole manuscript

2.

A minor shortcoming of the abstract is that the authors did not list keywords alphabetically; it is also questionable why the keywords start with a capital letter?

Thank you for your comment.

The abstract has been improved. Also, the keywords have been listed alphabetically and written in small letters based on your suggestion.

Abstract

Keywords.

3

The introduction lacks a clearly defined main goal and secondary goals.

Thank you for your valuable feedback.

The main objective have been stated at the end of the introduction.

To explore the mediating roles of OI and OC in these relationships between the independent variable (CSR) and the dependent variables (EP and TI) in this banking context.

To broaden the theoretical and empirical underpinnings to grasp the essence of the relationship between CSR, EP, TI, and the mediating factors of OI and OC

The main question of the current study:
How do organizational identification (OI) and commitment (OC) mediate the relation-ship between corporate social responsibility, employee performance, and turnover in-tention among employees in the banking sector?

Introduction

4

I appreciate the setting of the hypotheses, but why aren't they set in the introduction and why are there 12 of them? This number is extremely high. Consider reducing the number of hypotheses and/or establishing research questions with subsequent unambiguous answers in the final chapter, which would fulfill the very meaning of their scientific work.

Thank you for your comment. However, following social science research, hypotheses typically appear at the end of the literature review section, as they are developed based on the preceding literature review. Consequently, it can be challenging to include them in the introduction. Additionally, we followed the journal's guidelines, which may have referenced previous publications within the same journal.

As for the number of hypotheses, we are unable to reduce them since they represent the research framework, questions, objectives, and problem statement. Moreover, we have already tested them in the results section.

The research question has indeed been provided at the end of the introduction.

5

The relatively short introduction lacks a definition of the importance of the topic as well as an explanation of basic terms.

Thank you for this important suggestion.

The definition of the study variables has been added. Please refer to the section after the research framework figure.

After figure 1

6

Within the framework of the economy, employee performance, and research topics related to human resources, it is necessary to outline the labor law issue and devote at least one paragraph to the legislative side between the manager and the employer, the relationship of the manager himself as the representative of the employer to the employees, an explanation of the concept of who is the manager, because without this theoretical part, it will not be possible to provide a comprehensive view of the researched matter.

Thus, there is a lack of definition of these issues from a legislative point of view, as they do:

Peráček T. & Kaššaj M. (2023). The influence of jurisprudence on the formation of relations between the manager and the limited liability company. Juridical Tribune. 13 (1), pp. 43-62. 10.24818/TBJ/2023/13/1.04

Raguel, MM & Odeku, KO. 2023. Critical analysis of the failure of labor law to adequately protect atypical workers and its impact on human rights and fair labor practice. JURIDICAL TRIBUNE-TRIBUNA JURIDICA 13 (1), pp.63-81, doi: 10.24818/TBJ/2023/13/1.05

Muhamet BINAKU. 2021. VIOLATIONS OF THE RIGHTS OF EMPLOYEES IN PUBLIC AND PRIVATE SECTOR IN KOSOVO. Perspectives of Law and Public Administration, 10, (SI), pp. 133-137

To increase the scientific value of the work, I recommend the authors to expand this number by, for example, the above-mentioned works.

Thank you for your important comments and suggestions. I appreciate your perspective on the importance of outlining the labor law issue and the legislative side of the relationship between the manager and the employer. Indeed, these aspects are crucial in the broader context of employee performance and human resources.

However, the primary focus of this study is to explore the association between Corporate Social Responsibility (CSR), employee performance, and turnover intention, with a particular emphasis on the moderating role of organizational identification and commitment. The legislative aspects, while important, are not directly related to the core objectives of this study.

Incorporating a detailed discussion on labor law issues and the legislative side of the manager-employer relationship would significantly broaden the scope of the paper and may detract from the main focus.

Nevertheless, I acknowledge the relevance of your suggestion. Therefore, I will suggest it as a potential area for future research. This way, we can acknowledge the importance of these issues without diverting from the main objectives of the current study.

6.3 Limitations of study

7

I also recommend supplementing the scientific study with at least one paragraph devoted to the issue of the perspective of further research on the topic in the future.

Thank you for your valuable feedback.

The suggestions for future research have been added. Please refer to the last paragraph at the conclusion.

6.3 Limitations of study

Reviewer 4 Report

The article discusses the determinants of turnover intention among bank employees, examining factors such as employee satisfaction, corporate social responsibility, organizational commitment, and their interplay, with a focus on understanding the mechanisms affecting employee performance and retention in the banking sector. While the topic of corporate social responsibility (CSR), employee performance, and turnover intention has been explored in previous research, the specific focus on how organizational identification and commitment mediate these relationships within the context of Saudi Arabia's banking sector suggests a unique perspective, which can contribute to the existing body of knowledge in this field.

There are several aspects that could be improved:

The Abstract should be more concise and to the point. It includes a lot of details that are better suited for the main body of the research paper. An abstract should provide a high-level summary.  It mentions that CSR is significant within Saudi Arabia's banking sector, but it lacks a brief introduction or context to explain why this is important or what specific issues the study is addressing. This context would help readers understand the significance of the research. It  states that all variables explained the variance of EP by 42% and TI by 28%, but it doesn't elaborate on the practical implications of these findings or the significance of these percentages. The abstract mentions that twelve hypotheses were assessed, but it doesn't provide any details about what these hypotheses were or the specific findings. Providing a brief summary of the most important findings would make the abstract more informative. The mention of "Q2 being above 0" is not clear without context. It would be helpful to briefly explain what "Q2" refers to and why it matters in the context of the study. It repeats the study's aim multiple times, which could be condensed into a single clear statement of the research question or objective. The abstract should conclude with a brief summary of the main findings and their implications for the field of study. A revised abstract should aim to provide a clear, concise, and informative overview of the study, including its context, objectives, and key findings, without going into unnecessary detail.

The Introduction is quite lengthy and includes a lot of information that could be condensed for clarity. There is some redundancy in the text. For example, the concept of turnover intention is explained twice, once at the beginning and then later on. While the introduction provides a lot of factual information, it lacks engagement with the reader. It could benefit from a more engaging and concise opening that immediately draws the reader's attention to the research problem. The citations in the text are not consistently formatted.  The transition between discussing turnover intention, employee performance, and organizational identification and commitment is somewhat abrupt. The text frequently uses acronyms such as CSR, EP, TI, OI, and OC. While acronyms can save space, they can also make the text less accessible to readers who are not familiar with the field. It's helpful to spell out acronyms the first time they are used. The introduction mentions multiple factors that influence the banking sector in Saudi Arabia, but it could benefit from specific examples or statistics to illustrate the significance of these factors. The introduction concludes by mentioning the research gap related to organizational identification (OI) and commitment (OC) as mediators between CSR, EP, and TI. However, it could be more explicit in stating the precise research question or hypothesis that the study aims to address.

The Literature Review lacks a clear organizational structure. It jumps between different relationships (CSR-EP, CSR-OI, CSR-OC, etc.) without a smooth transition, making it challenging for the reader to follow the logical flow of the argument. There is some repetition in the literature review, with similar points and findings mentioned multiple times in different sections. This redundancy can be reduced to make the review more concise. Some paragraphs are overly lengthy and could be condensed for clarity and brevity.  The literature review provides summaries of various studies but lacks an integrated analysis that connects these findings and explains their significance collectively. The literature review primarily presents previous research findings but does not critically evaluate the strengths, limitations, or potential biases of these studies. Including a critical analysis of the existing literature would add depth to the review. While the literature review highlights the existing research on the relationships under investigation, it could more explicitly articulate the research gap that the current study aims to address. What specific aspects of the literature remain unexplored or require further investigation? The review uses terms like "organizational commitment," "organizational identification," and "corporate social responsibility" without providing clear definitions or explanations. It's essential to define these key terms to ensure reader comprehension. The hypotheses are introduced but could be more clearly stated and tied back to the literature. A brief explanation of why each hypothesis is proposed based on the literature would be helpful. While the discussion on organizational identification's mediating role is valuable, it's important to contextualize why this is relevant to the study's specific research questions and objectives. How does this mediation relate to the research problem being addressed? The section could benefit from a more seamless integration with the overall conceptual framework. Explain how organizational identification fits into the larger framework shown in Figure 1 and how it relates to other variables and hypotheses. The section provides a brief overview of Social Identity Theory, but it could be enhanced by offering a more detailed explanation of how this theory specifically applies to the study's context and variables. Some sentences, such as "The degree to which an individual gives importance to moral principles in their sense of who they are is what is meant by the term 'moral identity,'" could be rephrased for greater clarity and precision. Discuss how the Social Identity Theory framework and the role of organizational identification align with the specific research questions and hypotheses of the study. Provide a bridge between theory and practice.

While the "Method" section mentions the use of validated instruments, it lacks detailed information about the specific questions or items used within each scale. Providing a brief sample of the items or questions would help readers understand the constructs being measured. It would be beneficial to provide a brief justification for the sample size of 550 employees. Why was this sample size chosen, and how does it relate to the research objectives and statistical power? It's mentioned that the survey was conducted between 2020 and 2021, but more specific dates or a timeline of data collection would add transparency and context to the study. While the section mentions the avoidance of common method variance by separating variables into different sections, it could benefit from a more detailed explanation of how this was achieved. What steps were taken to minimize this potential issue? Although it's mentioned that a high response rate was achieved, providing the actual response rate (i.e., the percentage of surveys returned out of those distributed) would be informative. Ethical Considerations: The section lacks any mention of ethical considerations, such as obtaining informed consent from participants or ethics committee approval, if applicable. Including this information is important for transparency. While the sources of the measurement scales are mentioned, it would be helpful to include specific citations or references for each scale to guide readers who want to learn more about the instruments used. Some sentences are long and complex, which may hinder comprehension. Consider breaking down complex sentences for greater clarity.

While the "Results" section mentions that common method bias was assessed, the results of this assessment are not clearly presented. It would be helpful to provide a summary of the findings related to common method bias and explain how this potential issue was addressed. The section mentions discriminant validity tests, such as the heterotrait-monotrait ratio (HTMT), Fornell-Larcker criteria, and cross-loadings. While the tables are included, the interpretation and discussion of these results are missing. Readers need to know whether discriminant validity was established and whether there were any concerns. The explanation of effect sizes (f-squared) could be clearer. The authors mention that certain variables have a "substantial" effect size, but it would be helpful to provide more context on what "substantial" means in terms of practical significance. Explain how these effect sizes relate to the research hypotheses and their practical implications. While the authors mention that R2 values are significant, it would be beneficial to provide more interpretation of what these values mean in the context of the study. Discuss the practical significance of the explained variance in your endogenous variables. In the mediation analysis, it is mentioned that all mediating hypotheses were validated. However, it would be helpful to provide a brief discussion of the implications of these mediation findings. How do these mediations contribute to the understanding of your research questions? While statistical significance is reported throughout the section, it's important to bridge the gap between statistical significance and practical significance. Explain the real-world implications of the findings and why they matter. Ensure that you specify the units of measurement for any numerical results presented in the tables or figures. The authors should maintain consistency in presenting and interpreting results. For example, ensure that they use a consistent number of decimal places in tables.

The Discussion could benefit from a deeper exploration of the practical implications of the findings. Explain why the observed relationships matter in a practical sense, both for organizations and employees. In H4, the discussion mentions that the findings were not statistically significant. It's important to explore why this might be the case and whether there are any potential explanations for the lack of significance. When discussing the mediation analyses (H9, H10, H11, H12), provide a more detailed explanation of the implications of mediation. How do these mediating variables (Organizational Identification and Organizational Commitment) influence the relationships between CSR, Employee Performance, and Turnover Intention? Ensure that the discussion is presented in a clear and concise manner. Some sentences are lengthy and could be broken down for easier comprehension. Consider revising the flow of the discussion to ensure a logical progression of ideas. For example, start by summarizing the main findings before delving into the discussion of each hypothesis. Conclude the discussion section by summarizing the key takeaways and providing practical recommendations for organizations. What can businesses learn from these findings, and how can they apply them to enhance CSR practices and employee outcomes?

The Conclusion could benefit from improved clarity and organization. Separate the theoretical contributions, practical implications, and limitations into distinct paragraphs or sections for better readability. While it mentions the importance of developing organizational identification and commitment, it could provide more specific and actionable recommendations for organizations. What steps can banks in Saudi Arabia take to achieve these goals? The limitation related to the cross-sectional design is mentioned, but it would be helpful to explicitly discuss the challenge of establishing causality. Suggest how future research could address this limitation, such as through longitudinal studies. The conclusion acknowledges that the results may not be easy to generalize to other industries or nations. However, it could be strengthened by discussing potential strategies or approaches for adapting the findings to different cultural settings or industries. While the research focuses on organizational identification and commitment as mediators, it mentions the importance of considering other possible mediators or moderators. Expand on this point by suggesting potential variables or factors that future research could explore in this context. The conclusion doesn't explicitly synthesize the key findings. Consider summarizing the main findings and their implications in a concise manner before discussing limitations. When discussing external validity, provide more context on how the research's results might be relevant or applicable to other cultural settings or fields of endeavor. Proofread the conclusion for grammar and style consistency to ensure it reads smoothly.

References. Ensure consistency in the citation style used throughout the references. Some references include volume and page numbers, while others do not. Some references have missing publication years.

Some parts of the article, particularly the abstract and introduction, could benefit from greater conciseness. There is occasional redundancy in the text, with concepts or terms being explained multiple times. The paper generally adheres to good grammar and sentence structure, but there are some lengthy and complex sentences that could be broken down for better clarity.  Overall, while the quality of English language is good, there are areas where improvements can enhance clarity, readability, and overall quality.

Author Response

Response to Reviewer 4.

The Association Between Corporate Social Responsibility, Employee Performance, and Turnover Intention Moderated by Organizational Identification and Commitment

Manuscript ID:  sustainability-2613930

----------------------------------------------------------------------------------------------------------------------------------

We are very thankful for your positive reviews, comments, suggestions, and time spent reviewing this article (sustainability-2613930). Below, we discuss all the issues raised by you. Below, we list the reviewer’s comments, followed by our responses. The changes are highlighted in red color in the revised manuscript:

Response to Comments from Reviewer 4:

No.

Comment

Response / Correction made

Location

1.

 The abstract should be more concise and to the point. It includes a lot of details that are better suited for the main body of the research paper. An abstract should provide a high-level summary.  It mentions that CSR is significant within Saudi Arabia's banking sector, but it lacks a brief introduction or context to explain why this is important or what specific issues the study is addressing. This context would help readers understand the significance of the research.

It  states that all variables explained the variance of EP by 42% and TI by 28%, but it doesn't elaborate on the practical implications of these findings or the significance of these percentages.

 The abstract mentions that twelve hypotheses were assessed, but it doesn't provide any details about what these hypotheses were or the specific findings.

Providing a brief summary of the most important findings would make the abstract more informative.

The mention of "Q2 being above 0" is not clear without context. It would be helpful to briefly explain what "Q2" refers to and why it matters in the context of the study.

 It repeats the study's aim multiple times, which could be condensed into a single clear statement of the research question or objective. The abstract should conclude with a brief summary of the main findings and their implications for the field of study. A revised abstract should aim to provide a clear, concise, and informative overview of the study, including its context, objectives, and key findings, without going into unnecessary detail.

Thank you for your Important comment. We have considered all comments, and the abstract has been improved accordingly.

-The importance of SCR has been added.

-Explain the significance of R2 that has been added to the abstract.

-Hypotheses finding was illustrated.

-Q2 explanation has been added.

As you suggested, the abstract concluded with a brief summary of the main findings and their implications for the field of study.

Abstract

2.

The Introduction is quite lengthy and includes a lot of information that could be condensed for clarity. There is some redundancy in the text.

For example, the concept of turnover intention is explained twice, once at the beginning and then later on. While the Introduction provides a lot of factual information, it lacks engagement with the reader. It could benefit from a more engaging and concise opening that immediately draws the reader's attention to the research problem.

The citations in the text are not consistently formatted.  

The transition between discussing turnover intention, employee performance, and organizational identification and commitment is somewhat abrupt. The text frequently uses acronyms such as CSR, EP, TI, OI, and OC. While acronyms can save space, they can also make the text less accessible to readers who are not familiar with the field. It's helpful to spell out acronyms the first time they are used.

The Introduction mentions multiple factors that influence the banking sector in Saudi Arabia, but it could benefit from specific examples or statistics to illustrate the significance of these factors.

The Introduction concludes by mentioning the research gap related to organizational identification (OI) and commitment (OC) as mediators between CSR, EP, and TI. However, it could be more explicit in stating the precise research question or hypothesis that the study aims to address.

Thank you for your important suggestions.

The introduction has been entirely improved accordingly.

-Unnecessary information was removed.

-it has shortened, and now It is more concise and focused on the study's problem.

-Citations were modified.

- The coherence was checked, and the transitions have been improved.

- The research question was added.

Introduction

Literature Review 

3

The Literature Review lacks a clear organizational structure. It jumps between different relationships (CSR-EP, CSR-OI, CSR-OC, etc.) without a smooth transition, making it challenging for the reader to follow the logical flow of the argument.

Thank you for your comments. However, in the current literature review, we have discussed each relationship independently. So you found it without smooth transitions. We have followed the same way in previous research in the same journal. At the same time, we improved the paragraphs under each relationship, as you suggested.

Literature review

4

There is some repetition in the literature review, with similar points and findings mentioned multiple times in different sections. This redundancy can be reduced to make the review more concise.

Thank you for your important comment. We have removed the repetitions in the literature review. Please let us know if you have more remarks.

5

Some paragraphs are overly lengthy and could be condensed for clarity and brevity.  

Thank you for your feedback. We have checked the paragraphs and summarised many of them, but we can not summarise many sections to keep them useful for readers. 

Literature review

6

The literature review provides summaries of various studies but lacks an integrated analysis that connects these findings and explains their significance collectively.

The literature review primarily presents previous research findings but does not critically evaluate the strengths, limitations, or potential biases of these studies. Including a critical analysis of the existing literature would add depth to the review.

We appreciate your suggestion and recognize the importance of a comprehensive critical review, but in this study, we aim to develop and test the hypotheses.

Thus, our initial literature review aimed to develop hypotheses following a hypothesis-driven approach.

7

While the literature review highlights the existing research on the relationships under investigation, it could more explicitly articulate the research gap that the current study aims to address. What specific aspects of the literature remain unexplored or require further investigation?

Thank you for your comment.

The main gap was mentioned in the introduction. Please refer to lines 61-64.

Also, at the end of each relationship literature review section, we have highlighted the specific gap, as you can see in the last sentences, before writing the hypotheses.

Introdaction and literature review.

8

The review uses terms like "organizational commitment," "organizational identification," and "corporate social responsibility" without providing clear definitions or explanations. It's essential to define these key terms to ensure reader comprehension. The hypotheses are introduced but could be more clearly stated and tied back to the literature.

Thank you for raising this important comment.

Definitions have been added after the research framework.

9

A brief explanation of why each hypothesis is proposed based on the literature would be helpful.

Thank you for your comment. Let me justify that:

Proposing hypotheses based on the literature review is crucial for several key reasons. Firstly, it ensures that our research is firmly grounded in existing knowledge. The literature review represents a comprehensive survey of prior research and insights on our research topic. By formulating hypotheses rooted in this existing body of knowledge, we build upon the collective wisdom of previous studies.

Secondly, hypotheses provide the framework for testable predictions. They serve as clear, empirically examinable statements about the relationships between variables. By crafting hypotheses, we establish a foundation for data collection and statistical analyses. This systematic approach allows us to determine whether our expectations align with the observed results, fostering scientific rigour and objectivity in our research.

10

While the discussion on organizational identification's mediating role is valuable, it's important to contextualize why this is relevant to the study's specific research questions and objectives. How does this mediation relate to the research problem being addressed? The section could benefit from a more seamless integration with the overall conceptual framework. Explain how organizational identification fits into the larger framework shown in Figure 1 and how it relates to other variables and hypotheses.

Thank you for your feedback. The choice of Organizational Commitment (OC) and Organizational Identification (OI) as mediation variables is crucial to our study's research questions and objectives for the following reasons:

1. Theoretical Foundation: Extensive research in organizational behavior and management has consistently highlighted the significance of OC and OI in shaping employee attitudes and behaviors, making them key factors in understanding how CSR impacts employees.

2. Mediating Mechanisms: As cited in our literature review, Prior studies have demonstrated that OC and OI serve as mediating mechanisms explaining the link between independent variables and organizational outcomes. This validates their role as mediators in our study.

3. Contextual Relevance: Given our focus on the banking sector in Saudi Arabia, where CSR is gaining prominence, understanding the mediating roles of OC and OI is particularly relevant. It allows us to explore how CSR initiatives affect employees' attitudes and behaviors in this context.

4. Alignment with Objectives: Our research aims to examine CSR's impact on Employee Performance (EP) and Turnover Intention (TI) in the banking sector. By including OC and OI as mediating variables, we can comprehensively understand how CSR influences these outcomes, aligning with our research objectives.

11

The section provides a brief overview of Social Identity Theory, but it could be enhanced by offering a more detailed explanation of how this theory specifically applies to the study's context and variables. Some sentences, such as "The degree to which an individual gives importance to moral principles in their sense of who they are is what is meant by the term 'moral identity,'" could be rephrased for greater clarity and precision. Discuss how the Social Identity Theory framework and the role of organizational identification align with the specific research questions and hypotheses of the study. Provide a bridge between theory and practice.

Thank you very much for your comment.

More explanation about how SIT explains this study's relationships was added to the manuscript.

For further clarification, Social Identity Theory can be applied to the context and variables of this study in the following ways: First, Organizational Identification: Social Identity Theory posits that individuals derive a portion of their self-concept from the groups to which they belong, including their organization. In this study, organizational identification is a critical variable. Employees who strongly identify with their organization are more likely to perceive themselves as integral members of that group. This identification can influence their attitudes and behaviors, including their performance and intentions related to turnover. Second, Organizational Commitment: Social Identity Theory suggests that when individuals identify strongly with an organization, they are more likely to be committed to its goals and values. In this study, organizational commitment is another central variable. Employees who identify with their organization and its CSR initiatives may display higher levels of commitment, which, in turn, can affect their performance positively and reduce their intentions of leaving the organization.

Method" 

12

 While the "Method" section mentions the use of validated instruments, it lacks detailed information about the specific questions or items used within each scale. Providing a brief sample of the items or questions would help readers understand the constructs being measured.

Thank you for your useful suggestions.

We have added all survey items in the supplementary file.

Supplementary file.

13

It would be beneficial to provide a brief justification for the sample size of 550 employees. Why was this sample size chosen, and how does it relate to the research objectives and statistical power?

Thank you for your comment.

The information about sample size determination has been added. The sample size of our study is more than the minimum required.

 The study determined the suitable sample size using Krejcie and Morgan [89] table for sample size determination. Given that the population of employees within the Saudi Arabian banking sector was reported as 47,181 by the Saudi Arabian Monetary Authority (SAMA) in 2020, the minimum required sample size was calculated to be 381.

3.1. Design and Procedures

14

It's mentioned that the survey was conducted between 2020 and 2021, but more specific dates or a timeline of data collection would add transparency and context to the study.

Thank you for your important feedback.

The data was collected from March 2020 to February 202.

3.1. Design and Procedures

15

While the section mentions the avoidance of common method variance by separating variables into different sections, it could benefit from a more detailed explanation of how this was achieved. What steps were taken to minimize this potential issue?

Thank you for your useful comment.

We took several steps to minimize common method variance in our study. First, we designed separate questionnaire sections for each variable, ensuring participants were not asked about all variables in a single section. Secondly,  To provide context and emphasize the relevance of questions, we included clear introductory statements at the beginning of each section. Additionally, we assured respondents of the anonymity and confidentiality of their responses to minimize social desirability bias. Finally, we employed statistical techniques like factor analysis and covariance matrix examination to detect and control for potential methodological bias in our data analysis. These measures collectively created a conceptual divide between independent, mediator, and dependent variables, reducing common method variance and enhancing the internal validity of our study.

3.1. Design and Procedures

16

Although it's mentioned that a high response rate was achieved, providing the actual response rate (i.e., the percentage of surveys returned out of those distributed) would be informative.

Thank you for your feedback. However, since the survey was collected personally and online, we can not provide the exit number of the survey response percentage, which was distributed and how many were returned. As you know, the online form can be sent once to the bank management. Thus, we have mentioned receiving a high response rate because our target was 381, while we collected 550.

17

Ethical Considerations: The section lacks any mention of ethical considerations, such as obtaining informed consent from participants or ethics committee approval, if applicable. Including this information is important for transparency.

Thank you for this important feedback.

We have added ethical information to the manuscript as below:

Ethically informed consent was obtained from all subjects involved in the study. They voluntarily agreed to participate in this study, and we guaranteed them the data's confidentiality.

3.2. Scales

18

While the sources of the measurement scales are mentioned, it would be helpful to include specific citations or references for each scale to guide readers who want to learn more about the instruments used.

Thank you for your comment.

All references of measurement scales were added to the manuscript. At the same time, we have attached all survey items in a supplementary file.

3.2. Scales

Results

19

While the "Results" section mentions that common method bias was assessed, the results of this assessment are not clearly presented. It would be helpful to provide a summary of the findings related to common method bias and explain how this potential issue was addressed.

Thank you for this valuable comment.

More explanations have been added to the results. Regarding the less common method variance in our study, the steps were addressed in section 3.1. Design and Procedures

The table presented in Table 1 displays the distribution of variances for the variables under consideration. When we examine these variables collectively, they account for only 37.649 percent of the total variation. This percentage falls below the commonly accepted threshold of 50% [100]. This outcome implies that these results are less likely to have been significantly affected by common method bias, which is typically a concern when a single method or source dominates the data collection process. In our case, since the explained variance is below 50%, it suggests that common method bias is less likely to be a major influence on our results.

4.1. Common method bise

20

The section mentions discriminant validity tests, such as the heterotrait-monotrait ratio (HTMT), Fornell-Larcker criteria, and cross-loadings. While the tables are included, the interpretation and discussion of these results are missing. Readers need to know whether discriminant validity was established and whether there were any concerns.

Thank you for raising this important comment.

The interpretation of these three tables was added.

 Overall, This comprehensive approach enhances our confidence that the constructs in our study are distinct and not unduly influenced by shared variance.

4.2.2. Discriminant Validity

21

The explanation of effect sizes (f-squared) could be clearer. The authors mention that certain variables have a "substantial" effect size, but it would be helpful to provide more context on what "substantial" means in terms of practical significance. Explain how these effect sizes relate to the research hypotheses and their practical implications.

Thank you for this valuable suggestion.

We have added the required explanation.

According to the findings shown in Table 6, the f-square impact size varied widely, going from 0.006 (considered minimal) for OI on TI to 0.761 (considered substantial) for CSR on OC, which refers to the size or strength of the relationship between these variables within the context of this study. When we have substantial effect sizes, that means the relationships or differences identified in this study are not only statistically significant but also hold practical significance in the context of this research. These findings can have real-world implications, influencing decision-making and strategies within organizations or relevant fields.

4.3.1. The Strength of the Structural Model as an Explanatory Tool

22

While the authors mention that R2 values are significant, it would be beneficial to provide more interpretation of what these values mean in the context of the study.

 Discuss the practical significance of the explained variance in your endogenous variables.

Thank you for your comment.

We have considered this comment and added the interpretation of the R2 results.

 Significant R² values indicate that the study model effectively explains variations in the endogenous variables, providing insights into the factors that influence these outcomes. The practical significance lies in the implications for decision-making and strategy development, as it highlights which variables have a substantial impact and should be prioritized for intervention or improvement efforts in the relevant context.

4.3.1. The Strength of the Structural Model as an Explanatory Tool

23

In the mediation analysis, it is mentioned that all mediating hypotheses were validated. However, it would be helpful to provide a brief discussion of the implications of these mediation findings. How do these mediations contribute to the understanding of your research questions?

Thank you for your comment.

Following your suggestion, a brief discussion has been added to clarify these mediation relationships.

Our validated mediation findings in the context of the banking sector have significant implications. They deepen our understanding of how corporate social responsibility (CSR) influences employee performance and turnover intention through specific pathways. For instance, the identification of organizational identification and commitment as moderators highlights their crucial roles in shaping these relationships. This has practical importance, guiding organizations in optimizing CSR strategies to enhance employee performance and reduce turnover intention. Additionally, these findings contribute to the theoretical framework of CSR by providing empirical support and insights into its impact on employee outcomes. Thus, they enrich our comprehension of this complex relationship and provide a basis for practical actions.

4.3.3. Mediation Analysis

24

While statistical significance is reported throughout the section, it's important to bridge the gap between statistical significance and practical significance. Explain the real-world implications of the findings and why they matter

Thank you for your feedback. We appreciate the opportunity to clarify the practical significance of our research.

Consequently, our study's statistically significant findings have practical implications, including the potential to enhance employee performance and reduce turnover. CSR can be a valuable tool for organizations to boost productivity and retain talent. Additionally, we emphasize the significance of organizational identification and commitment as moderating factors, offering insights for tailoring CSR strategies to improve employee engagement. Overall, our research provides actionable guidance for strategic decision-making within organizations and underscores the practical relevance of our work.

4.3.3. Mediation Analysis

25

Ensure that you specify the units of measurement for any numerical results presented in the tables or figures.

Thank you for your useful comment.

It has been checked.

Results section

26

The authors should maintain consistency in presenting and interpreting results. For example, ensure that they use a consistent number of decimal places in tables

Thank you for your useful comment.

It has been checked and modified.

Results section

Discussion

27

The Discussion could benefit from a deeper exploration of the practical implications of the findings. Explain why the observed relationships matter in a practical sense, both for organizations and employees

Thank you for your comment. However, the practical contributions have been discussed in the conclusion section. We believe writing it here is not the appropriate place.

Conclusion

28

In H4, the discussion mentions that the findings were not statistically significant. It's important to explore why this might be the case and whether there are any potential explanations for the lack of significance.

Thank you for raising this important comment.

Actually, there was a tipo error. This relationship has a negatively significant impact.

Discussion

29

When discussing the mediation analyses (H9, H10, H11, H12), provide a more detailed explanation of the implications of mediation. How do these mediating variables (Organizational Identification and Organizational Commitment) influence the relationships between CSR, Employee Performance, and Turnover Intention?

Thank you very much for this valuable feedback.

The implication of mediation has been added to the discussion section based on your suggestions.

Discussion

30

Ensure that the discussion is presented in a clear and concise manner. Some sentences are lengthy and could be broken down for easier comprehension.

Thank you for your observation.

The discussion section has been checked carefully, and the long sentences were rewritten.

Discussion

31

Consider revising the flow of the discussion to ensure a logical progression of ideas. For example, start by summarizing the main findings before delving into the discussion of each hypothesis.

Thank you for this useful comment.

We have added a short introduction in the discussion section.

This research endeavor sought to unravel the multifaceted impact of Corporate Social Responsibility (CSR) on key organizational facets, including Organizational Identification, Organizational Commitment, Employee Performance, and Turnover Intention. In the forthcoming discussion, this study scrutinizes the findings through various lenses, drawing upon previous literature and culminating in practical recommendations for organizations. By exploring these dimensions, we aim to comprehensively understand the intricate relationship between CSR, employee performance, and turnover intention.

Discussion

32

Conclude the discussion section by summarizing the key takeaways and providing practical recommendations for organizations. What can businesses learn from these findings, and how can they apply them to enhance CSR practices and employee outcomes?

Really this a very helpful comment.

The conclusion of the discussion section has been added. Moreover, the practical recommendations are added at the discussion section's end.

Discussion

Conclusion

33

Separate the theoretical contributions, practical implications, and limitations into distinct paragraphs or sections for better readability.

Thank you for your comment.

We have separated the conclusion based on your suggestion.

Conclusion

34

While it mentions the importance of developing organizational identification and commitment, it could provide more specific and actionable recommendations for organizations. What steps can banks in Saudi Arabia take to achieve these goals?

Thank you for your comment. Based on your previous suggestion (32), we added the recommendations at the end of the discussion section. Thus, we think there will be repetition if we write it again here.

To translate these findings into practical recommendations for organizations, it is crucial to prioritize CSR initiatives that align with the organization's values and mis-sion. Actively involving employees in CSR activities and emphasizing their engage-ment can strengthen their sense of identification with the organization. Promoting a culture of commitment, valuing employee contributions, and ensuring CSR efforts res-onate with employees can enhance organizational commitment. Additionally, organi-zations should maintain transparency and consistency in CSR efforts to address per-ceptions of hypocrisy that can negatively impact employee attitudes. Regular meas-urement and monitoring of the impact of CSR initiatives on employee performance, identification, and commitment are essential for refining and improving CSR strate-gies. By implementing these recommendations, organizations can enhance their CSR practices and cultivate a more engaged and committed workforce, ultimately leading to improved organizational performance and reduced turnover rates.

End of the discussion

35

The limitation related to the cross-sectional design is mentioned, but it would be helpful to explicitly discuss the challenge of establishing causality. Suggest how future research could address this limitation, such as through longitudinal studies.

Thank you for your suggestion.

A longitudinal study has been added to the limitations as a suggestion for future consideration.

6.3 Limitations of study

36

The conclusion acknowledges that the results may not be easy to generalize to other industries or nations. However, it could be strengthened by discussing potential strategies or approaches for adapting the findings to different cultural settings or industries.

Thank you for your important comment.

We have suggested strategies that will help compare our findings with another industry or country.

6.3 Limitations of study

37

While the research focuses on organizational identification and commitment as mediators, it mentions the importance of considering other possible mediators or moderators. Expand on this point by suggesting potential variables or factors that future research could explore in this context.

Thank you for raising this important suggestion.

We have added many potential variables that future researchers can consider.

6.3 Limitations of study

38

The conclusion doesn't explicitly synthesize the key findings. Consider summarizing the main findings and their implications in a concise manner before discussing limitations.

Thank you for your comment. The key findings have been mentioned at the beginning of the conclusion section.

6. Conclusion

39

. Proofread the conclusion for grammar and style consistency to ensure it reads smoothly.

Thank you for this useful comment. We have checked the language for the conclusion and the whole manuscript by an expert.

Whole manuscript

40

References. Ensure consistency in the citation style used throughout the references. Some references include volume and page numbers, while others do not. Some references have missing publication years.

Thank you for your important feedback.

References have been checked carefully. Missing years and page numbers have been included.

References

41

Some parts of the article, particularly the abstract and Introduction, could benefit from greater conciseness. There is occasional redundancy in the text, with concepts or terms being explained multiple times. The paper generally adheres to good grammar and sentence structure, but there are some lengthy and complex sentences that could be broken down for better clarity.  Overall, while the quality of English language is good, there are areas where improvements can enhance clarity, readability, and overall quality

Thank you so much for all the constructive feedback. The whole manuscript has been checked in terms of the language by an expert.

Whole manuscript.

Round 2

Reviewer 2 Report

The manuscript has been improved and can be considered for publication 

Reviewer 3 Report

I am glad that the authors accepted my comments and I recommend publishing the article in this modified form.

Reviewer 4 Report

I am satisfied with the improvements that have been implemented in the article.